# Self-Supervised Learning on Molecular Graphs: A Systematic Investigation of Masking Design

**Jiannan Yang**  *jiannan.yang@stonybrook.edu*
*Stony Brook University*

**Veronika Thost**  *veronika.thost@ibm.com*
*MIT-IBM Watson AI Lab*

**Tengfei Ma**  *tengfei.ma@stonybrook.edu*
*Stony Brook University*

**Reviewed on OpenReview:** *https://openreview.net/forum?id=TE4vcYWRcc*

## Abstract

Self-supervised learning (SSL) plays a central role in molecular representation learning. Yet, many recent innovations in masking-based pretraining are introduced as heuristics and lack principled evaluation, obscuring which design choices are genuinely effective. This work cast the entire pretrain–finetune workflow into a unified probabilistic framework, enabling a transparent comparison and deeper understanding of masking strategies. Building on this formalism, we conduct a controlled study of three core design dimensions: masking distribution, prediction target, and encoder architecture, under rigorously controlled settings. We further employ information-theoretic measures to assess the informativeness of pretraining signals and connect them to empirically benchmarked downstream performance. Our findings reveal a surprising insight: sophisticated masking distributions offer no consistent benefit over uniform sampling for common node-level prediction tasks. Instead, the choice of prediction target and its synergy with the encoder architecture are far more critical. Specifically, shifting to semantically richer targets yields substantial downstream improvements, particularly when paired with expressive Graph Transformer encoders. These insights offer practical guidance for developing more effective SSL methods for molecular graphs.

## 1 Introduction

Graph neural networks (GNNs) have gained significant traction in chemistry due to their intrinsic compatibility with molecular graph structures (Duvenaud et al., 2015; Gilmer et al., 2017). A key challenge in this domain is that obtaining molecular property labels often requires specialized and costly experimental procedures (Ramakrishnan et al., 2014; Wu et al., 2018), which inherently limits the scale of empirically labeled datasets and hinders the rapid exploration of the vast chemical space. Powerful computational models are essential for exploring vast chemical spaces and accurately predicting molecular properties at scale. To reduce the need for extensive experimental labeling, researchers have increasingly adopted self-supervised learning (SSL) (Dara et al., 2022). SSL leverages supervisory signals from abundant unlabeled molecular data to pre-train models that can learn generalizable representations. These SSL methods for molecular GNNs are broadly categorized into two paradigms: masking-based pretraining (Hu et al., 2019; Hou et al., 2022) and contrastive learning (You et al., 2020; Liu et al., 2021a). The former involves masking attributes of sampled nodes or edges within a molecular graph and training the model to recover this hidden information, often using the original atom or bond properties as supervisory signals. The latter employs graph augmentations to generate positive and negative molecular pairs for contrastive learning. Both approaches aim to maximize the extraction of chemically relevant information from molecular structures, thereby improving the inductive

bias of GNNs for downstream tasks such as molecular screening and drug discovery, where labeled data is scarce.

This work focuses on the masking-based pretraining paradigm. A seminal work in this direction is Hu et al. (2019), which pioneered the use of graph neural networks (GNNs) with a masked prediction objective for molecular representation learning, demonstrating the effectiveness of reconstructing masked node or edge features. This work laid the foundation for subsequent studies. Over the years, various modifications have been introduced. These innovations can be broadly categorized along three main axes: (1) model architectures, such as adopting alternative GNN encoders or reconfiguring the overall learning framework (Rong et al., 2020; Hou et al., 2022; Liu et al., 2023); (2) masking distributions, involving novel strategies for selecting which parts of the graph to mask (Liu et al., 2024; Inae et al., 2024); and (3) prediction targets, which alter the nature of the information the model aims to reconstruct during pretraining (Xia et al., 2023; Yang et al., 2024).

While new studies often claim to surpass prior methods on benchmark datasets, our comprehensive evaluations demonstrated that many modifications to masking strategies do not yield significant performance gains when evaluated under more rigorously controlled settings. For instance, we find that replacing simple uniform sampling with more sophisticated distributions offers no consistent advantage. Furthermore, as noted by Koo & Kwon (2025), methodical comparisons that isolate the efficacy of specific masking strategies from other confounding factors remain limited. This makes it challenging to ascertain which design choices are genuinely effective. To address these ambiguities and provide a clearer understanding of masking-based SSL in molecular graphs, our contributions are as follows:

1. We formalize the masking-based pretraining pipeline for molecular graphs, factoring it into key design dimensions: masking distribution, prediction target, and encoder architecture. This enables a structured categorization and comparison of existing and novel approaches.

2. We conduct a rigorous comparative study by meticulously controlling experimental variables across these dimensions and hyperparameters, thereby isolating the true impact of different masking strategies and architectural choices on downstream task performance.

3. We introduce a model-agnostic information-theoretic analysis, using mutual information and Jensen-Shannon Divergence, to quantify the alignment between pretraining proxy tasks and downstream molecular property prediction. This analysis provides deeper insights into the underlying mechanisms driving observed performance differences.

## 2 Related Works

Self-supervised learning (SSL) has become a pivotal paradigm for learning general-purpose representations from large-scale unlabeled data (Jing & Tian, 2020; Wu et al., 2020; Liu et al., 2021b). In the graph domain, SSL methods are broadly categorized into two main paradigms. Contrastive learning (CL) learns discriminative representations by maximizing the agreement between different augmented views of a graph (You et al., 2020; Liu et al., 2021a; Wang et al., 2022). In parallel, a generative approach, often termed Masked Graph Modeling (MGM), learns by corrupting parts of the input graph and training a model to reconstruct the original information.

The underlying principle of MGM, learning representations by reconstructing masked portions of the input, was first popularized in natural language processing by models like BERT (Devlin et al., 2019). This powerful self-supervised paradigm was subsequently and concurrently adapted to other domains, including computer vision with Masked Image Modeling (He et al., 2022) and, central to this paper, the molecular domain. Within molecular learning, the masking principle has been applied across diverse data modalities of molecules. For instance, SMILES-BERT (Wang et al., 2019) treats molecules as 1D SMILES sequences and apply BERT-style token masking, directly leveraging advancements from NLP. Recent works such as MOAT (Long et al., 2024) and SMI-Editor (Zheng et al., 2025) have further broadened the design space by transforming molecules into multi-granularity prompts and integrating large language models.

In the 2D visual domain, MaskMol (Cheng et al., 2024) explores 2D molecular images, performing knowledge-guided pixel masking on atoms or functional groups to address specific challenges like activity cliffs. Furthermore, EMPP (An et al., 2025), physics-informed direction operates on 3D geometric structures, proposing to mask atomic positions and train equivariant GNNs to predict them, thereby learning about intramolecular forces. While each modality offers unique research directions, our work focuses on a principled analysis of masking design choices specifically within the prevalent 2D molecular graph paradigm.

## 2.1 Evolving Designs in Masked Modeling for 2D Molecular Graphs

The central architectural component for processing 2D molecular graphs is the Graph Neural Network (GNN), which serves as a powerful encoder that learns representations by operating directly on the graph topology and features. Applying masked modeling to these GNN-based systems began with the foundational framework of AttrMask (Hu et al., 2019), which masks atom or bond attributes and uses a simple MLP for reconstruction. Subsequent research has evolved this paradigm in multiple directions. Architecturally, GraphMAE (Hou et al., 2022), uses more expressive decoder and introducing mechanisms like re-masking in the latent space. Concurrently, more powerful encoder backbones like Graph Transformers were also leveraged to better model long-range dependencies (Rong et al., 2020; Liu et al., 2023; Yang et al., 2024). Beyond standard node-centric homogeneous graphs, some works have even explored fundamentally different input representations, such as the heterogeneous atom-bond graphs in MGMAE (Feng et al., 2022), as well as edge-centric architectures such as ESA (Buterez et al., 2025), which treat edges as primitives and interleave masked and vanilla self-attention based on edge adjacency, providing an alternative to node-focused masking paradigms.

Innovation has also occurred in the masking strategy itself. Research has moved from simple uniform random masking to adaptive distributions based on graph heuristics or learnable scorers to identify structurally important nodes (Liu et al., 2024). The masking granularity has also been a focus, with a clear trend towards higher-level semantic units, such as masking entire chemically meaningful motifs (subgraphs) instead of individual nodes (Zhang et al., 2021; Wu et al., 2023; Inae et al., 2024).

Another active research direction involves designing more semantically rich prediction targets. These include predicting a discrete index from a vocabulary representing structural subgraphs (Ma et al., 2024) or learned codebooks (Xia et al., 2023), as well as predicting pre-defined motif labels (Yang et al., 2024). Other approaches have also explored predicting high-dimensional continuous vector representations of local neighborhoods (Liu et al., 2023).

## 2.2 Current Challenges and The Need for Systematic Investigation

Despite the rapid proliferation of MGM methods on molecules, most studies focus on proposing a novel model and demonstrating its superiority on specific benchmarks, leading to several challenges: a lack of systematic analysis of the interplay between different design choices, a scarcity of controlled comparisons, and a limited understanding of the mechanisms behind observed performance differences. This is exacerbated by the trend of creating complex, hybrid frameworks that combine different SSL paradigms. For instance, works like GCMAE (Wang et al., 2024) and UGMAE (Tian et al., 2024) explicitly combine multiple SSL paradigms and introduce numerous components and loss terms. While powerful, the complexity of such models makes it increasingly difficult to attribute performance gains to specific design choices.

This highlights the urgent need for systematic investigation. Some prior work has started this process. The study by Koo & Kwon (2025) provided a comprehensive analysis of several lower-level masking design aspects, such as the masking phase (pretraining vs. fine-tuning), granularity (e.g., node vs. subgraph), location (feature vs. embedding), and key hyperparameters like masking ratio. While valuable, their analysis was conducted within a single architectural framework and did not cover higher-level design choices. Other works, like that of Wang et al. (2023); Cintas et al. (2023), have proposed new evaluation methodologies to characterize pre-trained representations beyond simple downstream task performance. These efforts reveal a core challenge: a more fundamental understanding of the causal links between pretraining design choices and the properties of the learned representations is required. Our work aims to address this gap by proposing a

formal probabilistic framework, conducting rigorously controlled experiments, and employing information-theoretic measures to provide deeper, more principled insights and practical guidance for the field.

## 3 Methodology

This section details the methodology for our systematic investigation. We begin by casting the pretrain-finetune pipeline into a unifying probabilistic framework (Sec. 3.1–3.2), allowing us to deconstruct and systematically compare masking strategies (Sec. 3.3). All molecular graphs are treated as undirected, non-singleton graphs.

### 3.1 Analysis Dimensions

While many existing works highlight the goal of pretraining as capturing the intrinsic chemical information embedded in molecular structures, they often lack a formal account of how the pretraining task relates to downstream property prediction. To provide a clearer perspective, we adopt a probabilistic model to describe the two-step pipeline commonly used in evaluating mask prediction approaches.

We consider the pretrain-finetune setting as applied to classification tasks. Formally, we define $X$, a random variable representing a label from a label space $\mathcal{L}$ that is assigned to a specific structural unit sampled from a space of possible units $\mathcal{S}(G)$ within a given graph $G = (V, E)$:

$$X : \mathcal{S}(G) \to \mathcal{L} \tag{1}$$

This general formulation is flexible and powerful. For instance, in simple node-level tasks, the space of structural units $\mathcal{S}$ is simply the set of nodes $V$. For higher-level tasks, $\mathcal{S}$ can be a family of subgraphs (e.g., chemical motifs).

Meanwhile, the graph-level (global) property used for downstream prediction is denoted as

$$Y : \mathcal{G} \to \{0, 1\} \tag{2}$$

where $\mathcal{G}$ is the space of all molecular graphs in a given downstream dataset.

Let $\mathcal{M}$ denote a specific masking strategy. This strategy defines how to sample masking indices $M \subset \{0, 1, \ldots, |\mathcal{S}| - 1\}$ for a graph $G$. The application of mask $M$ to $G$ from the dataset $\mathcal{G}_{\text{data}}$ yields the prediction target, the vector of true labels $X(\mathcal{S}_M)$, and the model's input, the masked graph $G_M$. Conventionally, $G_M$ is obtained by replacing the fundamental elements of $G$, such as node or edge features, to a fixed, non-existent vector $\mathbf{m}$ of the same dimension. Using this notation, we can frame the entire pipeline in the following way.

1. In the pretraining stage, the objective is to learn an optimal encoder–decoder pair $(f_\theta, g_\phi)$. Within the scope of our study, $f_\theta$ is a GNN, and the decoder $g_\phi$ is typically an MLP or another GNN. The objective is to minimize the expected loss for masked label prediction

$$\min_{\phi, \theta} \mathbb{E}_{G \sim \mathcal{G}_{\text{data}}, M \sim P_{\mathcal{M}}(\cdot | G)} \left[ L\left( g_\phi(f_\theta(G_M)), X(\mathcal{S}_M) \right) \right] \tag{3}$$

   We define the overall loss $L$ for a masked set $V_M$ as the mean of a loss function $\ell$ (e.g., cross-entropy) over all nodes in that set:

$$L\left( g_\phi(f_\theta(G_M)), X(\mathcal{S}_M) \right) = \frac{1}{|\mathcal{S}_M|} \sum_{u \in \mathcal{S}_M} \ell\left( g_\phi(f_\theta(G_M))_u, X(u) \right)$$

2. At test time, the pretrained encoder $f_{\bar{\theta}}$ is then used to initialize the backbone of the downstream classification model, which is further fine-tuned by minimizing:

$$\min_{\psi, \theta} \mathbb{E}_{(G,Y) \sim P(\cdot, \cdot)} \left[ \ell\left( h_\psi(f_\theta(G)), Y \right) \right] \tag{4}$$

   to improve prediction performance on unseen molecular graphs. Here $P(\cdot, \cdot)$ represents the joint distribution of graph $G$ and graph-level label $Y$ in the downstream dataset, and $h_\psi$ denotes the MLP classifier of graph-level properties.

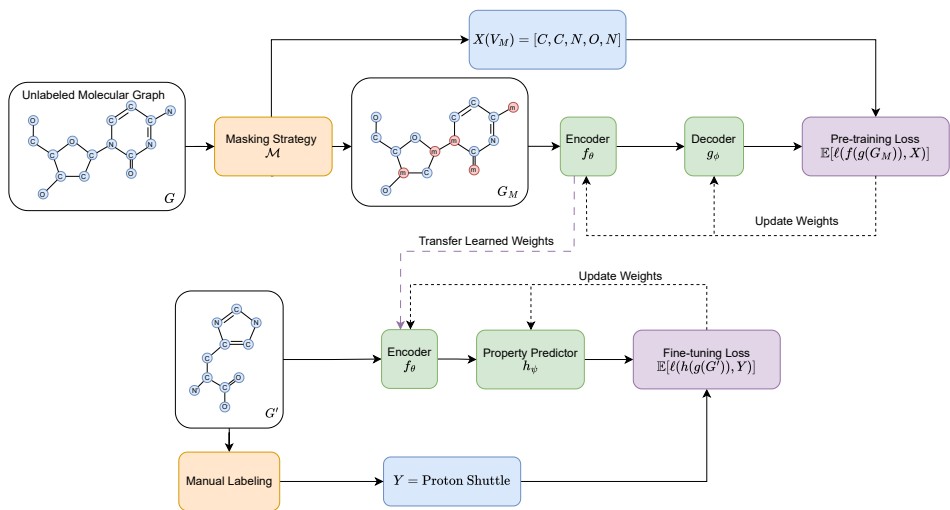

Figure 1: An Example of the Pretrain-Finetune Pipeline: Node Attribute Recovery

Based on this formulation, it becomes clear that design choices made during the mask prediction pretraining stage (Eq. 3) can be broadly factored into three main categories, as described below.

### 3.1.1 Masking Distribution

This first design dimension concerns the choice of the masking strategy, $\mathcal{M}$. Different strategies define different distribution to sample the masked nodes, which is why we refer this dimension as the **masking distribution**. In principle, the choice of $\mathcal{M}$ directly determines the marginal distribution, $P_{\mathcal{M}}(X|G)$, of the predicted labels $X$.

- **Uniform masking:** The sampling distribution is uniform over the set of nodes $V$, where each node is selected for masking with an equal and independent probability.

- **Heuristic masking:** The sampling distribution is determined by pre-defined heuristics based on graph structure.

- **Learnable masking:** The sampling distribution is dynamically learned during pretraining. This typically involves dedicated neural modules, that predict node importance scores, which are then used to parameterize the sampling probabilities.

### 3.1.2 Prediction Target

Another key design axis is to redefine the prediction target $X$ itself. In Figure 1, we adopted the formulation $X : V \to \{1, 2, \ldots, n\}$, representing a random variable mapping from nodes to discrete label values. This formulation can be altered in several ways, for instance, changing the sample space $V$, modifying the label range, or redefining the correspondence between the two.

One important example of such an alteration is to define $X$ as a mapping from subsets of nodes,

$$X : \mathcal{F}(V) \to \{1, 2, \ldots, m\}$$

where $\mathcal{F}(V)$ denotes a family of node subsets. This allows us to incorporate higher-level semantic labels, such as subgraph-level supervision.

In our experiments, we compare several classes of prediction targets. The specific methods corresponding to these classes will be detailed in Section 3.3.

- **Atomic Attribute Prediction:** The target is to reconstruct the original, low-level attributes of individual masked atoms, such as atom type or formal charge. This represents the most direct form of feature recovery.

- **Learned Node-level Token Prediction:** The target is a discrete token representing a learned, abstract representation of an atom. These tokens are typically derived from a separate, pretrained model like a vector-quantized encoder.

- **Structural Motif Prediction:** The target is a label corresponding to a higher-level, chemically meaningful substructure (i.e., a motif/subgraph) to which the masked atoms belong. This shifts the prediction from local atomic properties to broader structural semantics.

### 3.1.3 Encoder Architecture

The final design dimension we investigate is the choice of the encoder architecture, $f_\theta$. The encoder's capacity to model different types of structural dependencies is crucial, as it is the component responsible for generating transferable representations. Our study focuses on comparing two dominant paradigms for graph-based molecular encoding:

- **Message Passing Neural Networks (MPNNs):** This class of models iteratively updates node representations by aggregating information from their local neighborhoods. Due to their strong inductive bias for graph-structured data and computational efficiency, MPNNs have become the standard backbone for a wide range of molecular property prediction tasks.

- **Graph Transformers:** These architectures enhance message passing networks by incorporating global attention mechanisms, allowing every node to attend to every other node in the graph. This enables the direct modeling of long-range dependencies, which is challenging for standard MPNNs. While more expressive in principle, whether this increased capacity translates to better performance in masking-based pretraining remains an open question evaluated in our study.

**Other Components** Beyond the three core design dimensions, a complete pretraining pipeline involves choices about several auxiliary components. These often include the specific architecture of the decoder (e.g., a simple MLP versus a GNN-based decoder), the formulation of the loss function (e.g., standard cross-entropy versus a scaled cosine error), and other techniques such as applying a re-masking step to the latent representations before decoding (Hou et al., 2022). When comparing the main design dimensions in our study, we adopt a consistent configuration for these auxiliary components to ensure a fair comparison. Results from additional ablations on these components are provided in Appendix A.1 for completeness.

## 3.2 Principled Criteria for Signal Informativeness

Two hypotheses naturally arise in this framework: an effective pretraining signal can be engineered by either **optimizing the masking distribution $P_\mathcal{M}$** or **enriching the prediction target $X$**. These hypotheses rest on a common intuition: under controlled settings, if pretraining on the chosen definition of $X$ is to provide more useful information for predicting $Y$, then $X$ should exhibit stronger statistical dependence with $Y$. To formalize this intuition, we employ mutual information between $X$ and $Y$ as a principled, model-agnostic measurement to compare different design choices.

### 3.2.1 Mutual Information

To this end, we propose using the **mutual information (MI)** between $X$ and $Y$ as an information-theoretic measure of alignment. The mutual information $I(X;Y)$ quantifies how much information about the global label $Y$ we can obtain by knowing the local label $X$, and is defined as[1]:

$$I(X;Y) = \sum_{x,y} P(x,y) \log \frac{P(x,y)}{P(x)P(y)} \tag{5}$$

---

[1]Formally, we define the sample space of $(X,Y)$ as $\Omega = \bigsqcup_{G \in \mathcal{G}} \mathcal{S}(G) = \bigcup_{G \in \mathcal{G}} \{(u,G) : u \in \mathcal{S}(G)\}$ so that each structural unit $u$ is paired with its parent graph $G$, ensuring the joint distribution is well-defined.

where $x$ ranges over node or motif labels, and $y$ is the graph-level label.

A higher value of $I(X;Y)$ indicates stronger statistical dependence between the sampled local label variable $X$ and the downstream property $Y$, suggesting that the pretraining signal may provide richer information about the target task. This formulation allows us to compare the potential informativeness of different prediction targets and masking distributions in a model-agnostic manner, independent of encoder architecture.

### 3.2.2 Analysis of Conditional Distributions for Low-Frequency Labels

While MI provides a holistic measure of dependence, its averaging nature can obscure important details. Specifically, the influence of highly discriminative but infrequent local labels might be diluted by more common ones. This is a particularly relevant concern when comparing prediction targets of different semantic levels (e.g., common atoms vs. rare functional groups).

To probe the discriminative power of local labels beyond this average effect, our analysis leverages the **Jensen-Shannon Divergence (JSD)**. The JSD allows for a direct comparison of the conditional label distributions, $P(X|Y=1)$ and $P(X|Y=0)$. Based on the hypothesis that impactful local labels are often infrequent, we focus our JSD analysis on a subset of low-frequency labels, $S_\tau$, defined as:

$$S_\tau = \{x \in \mathcal{X} \mid P(x) < \tau\} \tag{6}$$

where $\mathcal{X}$ is the set of all unique local labels, $P(x)$ is the empirical probability of label $x$, and $\tau \in (0, 1]$ is a probability threshold. We then estimate the conditional probability distributions restricted to this subset, $P(X|Y=y, S_\tau)$, as follows:

$$P(X = x|Y = y, S_\tau) = \frac{N(X = x, Y = y)}{\sum_{x' \in S_\tau} N(X = x', Y = y)} \tag{7}$$

where $N(X = x, Y = y)$ is the count of occurrences of label $x$ in graphs of class $y$. The JSD is then computed between $P(X|Y=1, S_\tau)$ and $P(X|Y=0, S_\tau)$ to evaluate how the distinguishability of rare labels varies across different prediction target types.

### 3.3 Instantiating the Design Dimensions

To ground our theoretical framework in practice, this section maps a set of reproduced pretraining methods to the design dimensions outlined in Section 3. Instead of a simple chronological review, we organize this section to mirror our framework's structure. We first introduce AttrMask as the foundational baseline in detail. Subsequent subsections then explore how various methods have innovated upon this baseline along each of the three primary axes: masking distribution, prediction target, and architectural components. This approach allows for a clear, dimension-wise comparison while presenting each method in a logical, dependency-aware order. A complete overview of all method configurations is provided in Appendix A.1, see Table 8.

**AttrMask.** The pioneering work of Hu et al. (2019) introduced Attribute Masking (AttrMask), which serves as the foundational baseline in our study. Within our probabilistic framework, AttrMask can be precisely defined by instantiating each core component. The **prediction target**

$$X_{\text{type}} : V \to \{0, 1, 2, \dots, 118\}$$

is the original atomic attribute (e.g., atom type[2]), where the space of structural units $\mathcal{S}(G)$ for each graph $G$ is the set of nodes $V$. The **masking distribution** $P_\mathcal{M}$ is a uniform distribution over these nodes, from which a subset $V_m \subset V$ is sampled. To create the corrupted graph $G_M$, the feature vectors $x_v$ of these selected nodes are replaced with a special mask token $\mathbf{m}$ of the same dimension. Finally, an **encoder** $f_\theta$, typically an MPNN, processes $G_M$ to produce node embeddings, and a simple MLP acts as the **decoder** $g_\phi$ to predict the original attributes $X(v)$ for $v \in V_m$ by minimizing a cross-entropy loss. See Figure 1.

---

[2]In the range of $X_{\text{atom}}$, 0 stands for the element class of unknown atoms.

**GraphMAE.** Building on the AttrMask framework, GraphMAE (Hou et al., 2022) introduces key architectural innovations to reformulate the task as a more complete *masked graph auto-encoder*. It retains the same uniform masking distribution and atomic attribute prediction target as AttrMask, but enhances the **architecture**, primarily in the decoding process. Its key contributions are:

(1) **A Re-masking step**, where the embeddings of masked nodes are *again* replaced by a special token before being passed to the decoder.

(2) **A GNN-based decoder**, where another GNN layer is used as part of the decoder $g_\phi$ to further process latent codes before a final MLP predicts the node attributes.

Additionally, GraphMAE proposes using a **scaled-cosine error (SCE) loss**, shown in Equation 8, instead of cross-entropy to down-weight easy-to-predict examples.

$$\mathcal{L}_{\text{SCE}} = \frac{1}{|V_m|} \left( 1 - \frac{x_i^T \tilde{x}_i}{\|x_i\| \cdot \|\tilde{x}_i\|} \right)^\gamma, \ \gamma \geq 1 \tag{8}$$

### 3.3.1 Innovations in Masking Distribution

The methods discussed so far, AttrMask and GraphMAE, both rely on a simple uniform distribution for selecting nodes to mask. However, the hypothesis that non-uniform, structure-aware masking distributions $(P_\mathcal{M})$ could provide a more effective pretraining signal has also been explored, with StructMAE (Liu et al., 2024) being a representative example. This approach builds upon the GraphMAE framework to introduce heuristic and learnable masking strategies.

**StructMAE.** StructMAE (Liu et al., 2024) extends GraphMAE by replacing uniform node sampling with structure-aware masking distributions. Two variants are proposed:

- **StructMAE-P** Nodes are ranked by PageRank scores, computed as $x^{(t+1)} = \alpha D^{-1} A x^{(t)} + (1 - \alpha)p$ until convergence. To avoid always masking the same high-ranked nodes, a perturbed top-$k$ selection is applied, where random noise is added to candidate scores and the effective mask rate is gradually annealed during training.

- **StructMAE-L.** Instead of heuristics, node importance scores are learned jointly with the encoder via a shallow GNN and MLP scorer. The same perturbed top-$k$ mechanism is then applied to these learned scores, enabling a flexible, data-driven masking distribution.

Both variants share the GraphMAE auto-encoding framework; only the masking distribution differs. The full algorithmic details are provided in Appendix A.12.

**MoAMa.** Motif-aware Attribute Masking strategy (MoAMa) (Inae et al., 2024) partitions a molecular graph into multiple connected subgraphs, or motifs, using the BRICS decomposition (Degen et al., 2008). Rather than uniformly sampling individual nodes, MoAMa samples at the motif level. It employs a non-adjacent motif selection policy. Specifically, this algorithm iteratively samples a motif from the pool of available candidates and then removes its direct neighbors from the pool for subsequent selections within the same graph. This principle is likely intended to prevent the creation of large, contiguous masked regions, which could sever information pathways for local message-passing encoders.

It is worth noting that the original MoAMa framework additionally incorporates a molecular fingerprint-based contrastive loss. However, since our study focuses solely on the masking strategy, for a fair comparison, we exclude the auxiliary loss from our implementation.

### 3.3.2 Innovations in Prediction Target

We now turn to the second major axis of design: the prediction target $X$ itself. The following methods move beyond reconstructing simple atomic attributes, as done in the previously discussed methods, by proposing

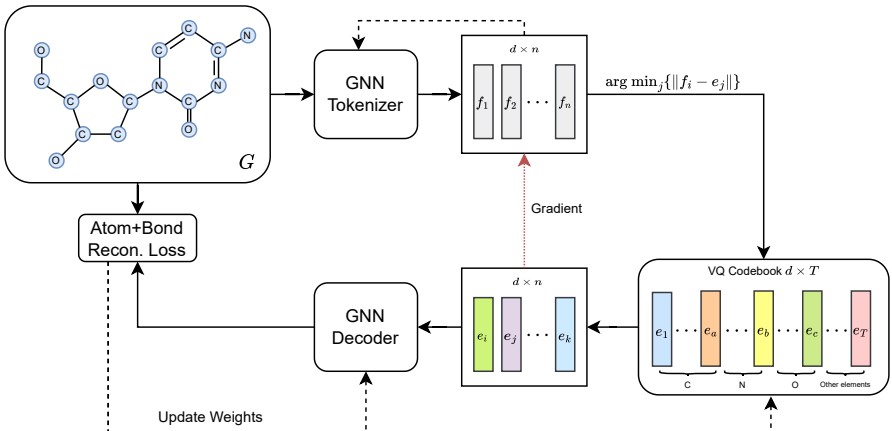

Figure 2: Group VQ-VQE Pretraining for Tokenizer and VQ Codebook in MAM

semantically richer targets. These include approaches that predict learned, abstract node representations or explicit, chemically-meaningful substructures.

**MAM.** Inspired by VQ-VAE (Van Den Oord et al., 2017), Xia et al. (2023) introduced Masked Atom Modeling (MAM) as an alternative to AttrMask, aiming to expand the prediction space beyond atomic types as part of their Mole-BERT framework. Specifically, it relies on a GNN tokenizer, denoted as $T_\varphi$, and a learned VQ codebook, $Q_\vartheta$, whose parameters are fixed after a separate pretraining phase (see Figure 2).

In our study, we consider two instantiations of MAM based on how node labels are assigned. These variants differ in whether the pretrained VQ codebook is explicitly used during masking prediction.

**(1) MAM-A (Argmax labeling):** A discrete pseudo-label is generated by applying an $\arg\max$ operation to the output of the GNN tokenizer $T_\varphi$:

$$X_A : V \to \{0, 1, \ldots, 511\}$$
$$v \mapsto \operatorname*{argmax}_i \{\mathcal{T}_\varphi(v)_i\}_{i=0}^{511}$$

This variant does not rely on the VQ codebook $Q_\vartheta$.

**(2) MAM-VQ (Codebook labeling)[3]:** Node labels are assigned by finding the index of the nearest entry in the VQ codebook $Q_\vartheta$ to the output of the tokenizer $T_\varphi$:

$$X_{VQ} : V \to \{0, 1, \ldots, 511\}$$
$$v \mapsto \operatorname*{argmin}_i \{\|\mathcal{T}_\varphi(v) - \mathcal{Q}_\vartheta[i, :]\|\}_{i=0}^{511}$$

The pretraining and subsequent use of the VQ codebook in MAM-VQ involves a nuanced two-stage process. For a detailed explanation, please refer to Appendix A.11.

**MotifPred.** Based on the idea of motif-level supervision in ReaCTMask (Yang et al., 2024), we propose a simplified motif prediction task, denoted as MotifPred. Specifically, we train the model to predict a unique pre-assigned label for each motif:

$$X_{motif} : \mathcal{F}(G) \to \mathcal{L}_{motif}$$

---

[3]We implemented MAM-VQ based on the original paper's description, as only the MAM-A variant is available in the official codebase.

where $\mathcal{F}(G)$ is the set of all motifs in $G$. To generate a prediction for a given motif, the final node representations of all atoms within that motif are first aggregated (e.g., via sum pooling) into a single motif-level representation. This aggregated vector is then used by the decoder to predict the corresponding motif label.

To create a manageable set of labels $\mathcal{L}$, we follow Yang et al. (2024) and pre-compute the motif decomposition using the refined BRICS algorithm (Zhang et al., 2021). This pre-computation not only helps decrease the vocabulary size $|\mathcal{L}|$, but also enables a more efficient sampling mechanism compared to MoAMa's on-the-fly approach. In contrast to MoAMa, MotifPred also masks only a subset of atoms within a selected motif rather than the entire substructure. This setting simplifies the original ReaCTMask, which was performed within a disjoint union of molecular graphs in a chemical equation. To ensure a controlled comparison across different design dimensions, we implement MotifPred using both GraphGPS (Rampášek et al., 2022) and message passing networks.

### 3.3.3   Encoder Architecture

The final design dimension in our framework is the choice of the encoder $f_\theta$, which dictates how structural information is processed and aggregated within the graph. Our study compares two GNN backbones used in the previous works, representing local and global information flow, respectively.

**Message Passing Neural Networks (MPNNs)**   The architectural backbone employed by most discussed methods in this study, is the Graph Isomorphism Network with Edge features (GINE) (Xu et al., 2019; Hu et al., 2019), a Message Passing Neural Network. The GINE layer updates a node's representation $h_v$ by aggregating features from its neighborhood $\mathcal{N}(v)$ according to the following rule for layer $k$:

$$h_v^{(k)} = \text{MLP}^{(k)} \left( (1 + \epsilon^{(k)}) \cdot h_v^{(k-1)} + \sum_{u \in \mathcal{N}(v)} \text{ReLU} \left( h_u^{(k-1)} + e_{u,v} \right) \right) \tag{9}$$

where $h_v^{(k-1)}$ is the representation of node $v$ from the previous layer, $e_{u,v}$ is the feature of the edge connecting nodes $u$ and $v$. This iterative, local aggregation provides a strong inductive bias for graph topology but inherently limits the model's receptive field.

**Graph Transformers**   To capture dependencies beyond local neighborhoods, we also employ a more expressive Graph Transformer, specifically GraphGPS (Rampášek et al., 2022). These architectures augment the message-passing framework with a global attention mechanism, enabling any node in the graph to directly attend to any other node. A conceptual representation of a GraphGPS layer's update for a node $v$ is:

$$h_v^{(k)} = h_v^{(k-1)} + \text{FFN}^{(k)} \left( \text{LocalMP}^{(k)}(h^{(k-1)})_v + \text{GlobalAttention}^{(k)}(h^{(k-1)})_v \right) \tag{10}$$

This capacity for modeling long-range dependencies makes them, in theory, better suited for pretraining tasks that require a global understanding of the graph. Indeed, this principle is demonstrated by ReaCT-Mask (Yang et al., 2024), which employs this transformer-based GNN encoder to enable information flow between the disconnected components (i.e., reactants and products) of a reaction graph. The empirical comparison of these two encoder types in Section 5 is therefore a key component of our investigation.

## 4   Experimental Protocol

We adopt a standardized two-stage protocol: (1) self-supervised pretraining on 2M molecules sampled from ZINC15 (Sterling & Irwin, 2015; Hu et al., 2019), and (2) fine-tuning and evaluation on 11 MoleculeNet benchmarks (Wu et al., 2018), with supplementary validation on curated datasets from Polaris (Wognum et al., 2024) (See A.5).

**Pretraining.**   We primarily compare two encoder backbones: GIN and GraphGPS, both implemented with edge-aware GINE layers (Hu et al., 2019). The hidden dimension is fixed to 300, trained for 100 epochs with Adam optimizer. To ensure fairness, mask ratios follow prior work but are aligned where needed: 0.15 for

AttrMask-family baselines, 0.25 for GraphMAE/StructMAE, and 0.30 (with 50% intra-motif atom masking) for MotifPred. All other hyperparameters (batch size, dropout, learning rate) are summarized in Table 1. The decoder is a single-layer MLP by default, except GraphMAE/StructMAE which adopt a GNN-based decoder (PReLU + linear + GIN layer), see A.8 for more details. Method-specific modules such as vector quantizers and masking scorers are documented in Table 2.

Table 1: Pretraining configuration of two backbone models.

| Component | GIN | GraphGPS |
|---|---|---|
| Encoder layers | 5 GIN layers | 5 GPS blocks |
| Hidden dimension | 300 | 300 |
| Dropout | 0.0 | 0.0 |
| Attention heads | – | 8 |
| Optimizer | Adam | |
| Learning rate | $1 \times 10^{-3}$ | |
| Batch size | 256 | 256 |
| Dropout rate | 0.0 | 0.0 (GIN) 0.5 (Attn) |
| Epochs | 100 | 100 |

Table 2: Additional components used by specific pretraining methods.

| Method | Component | Key Configuration |
|---|---|---|
| MAM | Vector Quantizer $\mathcal{Q}_\vartheta$ | Codebook size 512, token dim 300, commitment cost 0.25. |
| | Tokenizer $\mathcal{T}_\varphi$ | 5-layer GIN with hidden dim 300; trained jointly with VQ codebook for 60 epochs. |
| StructMAE-L | Node Importance Scorer | 2-layer MLP and 1-layer GIN (each with input/output dim 300); their outputs are aggregated and pooled to produce scalar node scores. |
| MotifPred | Motif-Atom Map | Pre-computed mapping between motifs and constituent atoms. |

**Fine-tuning.** For downstream tasks, we attach a linear prediction head and fine-tune the encoder using scaffold-based 8:1:1 splits. Classification and regression tasks follow the settings in Table 3. To reduce variance, all experiments are repeated with 5 random seeds.

Table 3: Fine-tuning configuration across task types.

| Parameter | Classification | Regression |
|---|---|---|
| Prediction head | Linear layer (input dim = 300) | |
| Optimizer | Adam | |
| Learning rate | $1 \times 10^{-3}$ | |
| Epochs | 100 | 100 |
| Dropout rate | 0.5 | 0.2 |
| Batch size | 32 | 256 |

**Method Mapping.** Finally, Table 4 categorizes the key baselines along the three design dimensions (masking distribution, prediction target, encoder). This provides a compact reference for how each method is positioned within our framework. Full implementation details are deferred to Table 8.

*Remark:* SupLearn denotes the randomly initialized model train on downstream labels from scratch. Methods with the suffix (T) indicate the encoder is GraphGPS; otherwise, GIN. Suffix -P indicates PageRank-based masking, and -L indicates learnable masking.

Table 4: Categorization of methods

| Method | Masking Dist. | Prediction Target | Encoder | Mask Ratio |
|--------|---------------|-------------------|---------|------------|
| SupLearn | – | – | GIN, GPS | – |
| AttrMask | Uniform | Atom Type | GIN, GPS | 0.15 |
| MAM-A | Uniform | Argmax Label | GIN, GPS | 0.15 |
| MAM-VQ | Uniform | VQ Code | GIN, GPS | 0.15 |
| MotifPred | Uniform | Motif Label | GIN, GPS | 0.15 |
| MoAMa | Intra-motif | Atom Type | GIN, GPS | 0.15 |
| GraphMAE | Uniform | Atom Type | GIN | 0.25 |
| StructMAE-P | PageRank | Atom Type | GIN | 0.25 |
| StructMAE-L | Learnable | Atom Type | GIN | 0.25 |

# 5 Experimental Results

## 5.1 Masking Distribution

We compare uniform, heuristic (PageRank), and learnable masking distributions across both classification and regression tasks. We additionally implemented heuristic and learnable variants (AttrMask-P/L, MotifPred-P/L); implementation details are in A.10.

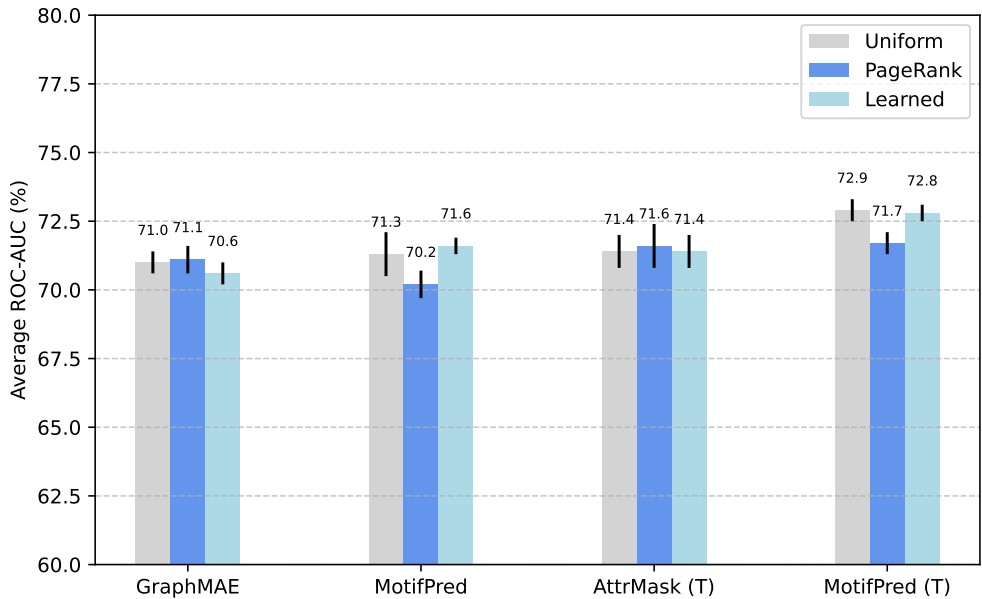

Figure 3: Effect of masking distributions on MoleculeNet classification

### 5.1.1 Discrete Molecular Properties

Figure 3 reports the average ROC-AUC (%), with error bars denote the standard deviations over 5 random seeds. Each group of bars corresponds to a specific pretraining method (GraphMAE, MotifPred, AttrMask (T), and MotifPred (T)), where the prediction target and model architecture are fixed, and only the masking distribution is varied. It shows that across all baselines, neither PageRank-based nor learnable masking consistently outperforms uniform sampling. This is not an artifact of fine-tuning, we also conducted linear probing experiments, where the pretrained encoder is frozen and only a linear head is trained (see Table 12 in A.1.2). The results confirm the same trend: non-uniform masking does not provide improvements over uniform masking.

We next examine the mutual information between the downstream label $Y$ and the masked label $X$ under each sampling strategy to further interpret these results.

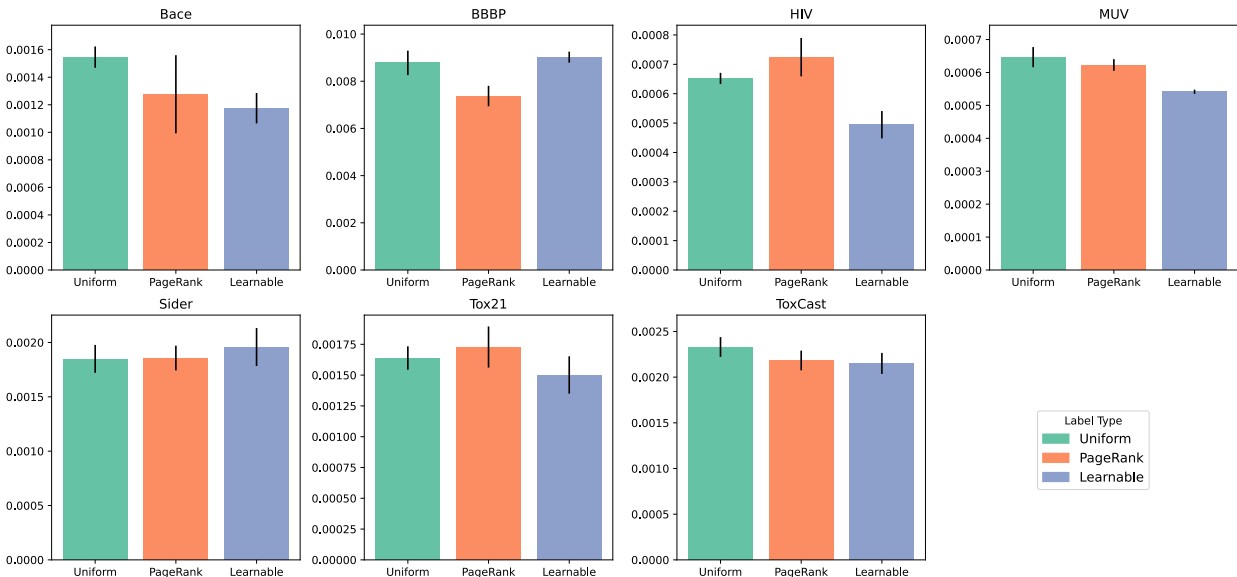

Figure 4: MI Between Sampled Atom Labels and Property Labels Across Different Masking Strategies

**Mutual Information Analysis** We compute the mutual information between $X$ and $Y$ under each sampling strategy. Here, $X$ is defined as the sampled **atom type** label, while the sampling distribution varies over the node set $V$. For each distribution, we sample $|V|$ nodes from every graph $G$, pair each sampled node label $x$ with the corresponding graph-level property label $y$, and estimate the joint distribution of $(X, Y)$. The mutual information is then computed separately for each classification dataset[4].

Each bar in Figure 4 shows the average MI over five random seeds, with error bars denoting standard deviation. Across datasets, MI scores under different masking strategies are very similar, with no consistent gain from PageRank-based or learnable masking. This indicates that structure-guided masking has little impact on the dependence between sampled atom labels $X$ and downstream labels $Y$. The analysis is model-agnostic and complements Figure 3, where prediction target and encoder are fixed. The consistently small MI variation across strategies explains why downstream performance remains stable despite changes in the sampling distribution.

### 5.1.2 Continuous Molecular Properties

Table 5 reports the RMSE on four datasets with regression tasks.

---

[4]For multi-task datasets with multiple graph-level labels (e.g., Sider, ToxCast), we compute MI w.r.t. a single task to ensure tractability.

|  | ESOL | Lipophilicity | Malaria | CEP | Average RMSE |
|---|---|---|---|---|---|
| # of data | 1,128 | 4,200 | 9,999 | 29,978 | - |
| SupLearn | 1.387 (0.087) | 0.796 (0.019) | 1.105 (0.011) | 1.341 (0.010) | 1.157 |
| GraphMAE | 1.195 (0.024) | 0.781 (0.011) | 1.116 (0.002) | 1.384 (0.016) | 1.119 |
| StructMAE-P | 1.195 (0.021) | 0.762 (0.011) | 1.119 (0.009) | 1.385 (0.016) | 1.115 |
| StructMAE-L | 1.310 (0.029) | 0.756 (0.015) | 1.111 (0.015) | 1.357 (0.007) | 1.134 |
| SupLearn (T) | 1.036 (0.084) | 0.744 (0.039) | 1.130 (0.007) | 1.689 (0.064) | 1.150 |
| AttrMask (T) | 1.194 (0.073) | 0.747 (0.015) | 1.105 (0.013) | 1.260 (0.027) | 1.077 |
| AttrMask-P (T) | 1.010 (0.043) | 0.693 (0.023) | 1.127 (0.016) | 1.482 (0.047) | 1.078 |
| AttrMask-L (T) | 1.166 (0.031) | 0.770 (0.018) | 1.127 (0.014) | 1.263 (0.043) | 1.082 |

Table 5: MoleculeNet: Regression Tasks (RMSE) over Different Masking Distribution

Across all evaluated models, including GraphMAE variants, and AttrMask (T) variants, we observe no consistent benefit from using structure-aware masking strategies. In all cases, performance remains comparable across sampling methods, with variations falling within the expected range of random training noise.

The results on continuous molecular property prediction reinforce our conclusion that modifying the masking distribution yields limited benefits. In particular, on Malaria, none of the pretrained variants outperform the supervised baseline trained from scratch. On CEP, while the Transformer-based methods show better performance than the supervised baselines, the benefit appears orthogonal to the choice of masking distribution.

Finally, beyond downstream performance and information-theoretic alignment, we also note the practical implications of computational cost. As quantified in our pre-training time comparison (see A.3 for details), sophisticated heuristic and learnable masking strategies introduce significant computational overhead, with some methods being 2-4x slower than the simple uniform baseline. This cost, combined with their lack of performance benefits, reinforces the practicality of uniform sampling.

## 5.2 Prediction Target

We next analyze how different prediction targets $X$ affect pretraining effectiveness, by examining their informativeness as supervision signals.

Table 6: Formal definitions of prediction targets used in pretraining.

| Target Type | Definition | Method Name |
|---|---|---|
| Atom Type | $X_{\text{type}} : V \to \{0, 1, \ldots, 118\}$ (Element class) | AttrMask |
| Argmax Label | $X_A : V \to \{0, 1, \ldots, 511\}, v \mapsto \arg\max_i \{\mathcal{T}_\varphi(v)_i\}_{i=0}^{511}$ | MAM-A |
| VQ Code | $X_{\text{VQ}} : V \to \{0, 1, \ldots, 511\}, v \mapsto \arg\min_i \{\|\mathcal{T}_\varphi(v) - \mathcal{Q}_\vartheta[i, :]\|\}_{i=0}^{511}$ | MAM-VQ |
| Motif Label | $X_{\text{motif}} : \mathcal{F}(V) \to \{1, \ldots, m\}$ (Motif class) | MotifPred |

### 5.2.1 Impact on Downstream Performance

The downstream performance of different pretraining targets is reported in Table 7. Full results with standard deviations are provided in Appendix A.1.

Table 7: Classification: ROC-AUC (↑); Regression: RMSE (↓). Best results per row are in **bold**.

(a) GIN Encoder

| Dataset | SupLearn | AttrMask | MAM-A | MAM-VQ | MotifPred |
|---|---|---|---|---|---|
| Tox21 | 73.9 | 75.8 | 74.9 | 75.7 | **76.6** |
| ToxCast | 63.6 | 64.3 | 61.7 | 63.3 | **64.5** |
| Sider | 57.7 | 60.2 | 58.2 | 59.4 | **60.5** |
| MUV | 73.1 | 72.3 | **77.8** | 76.0 | 76.8 |
| HIV | 74.3 | 76.5 | 76.8 | **76.9** | 76.8 |
| BBBP | **67.7** | 63.4 | 65.4 | 64.6 | 64.7 |
| Bace | 68.8 | 78.0 | **80.9** | 78.1 | 79.3 |
| *Average* | 68.4 | 70.1 | 70.8 | 70.6 | **71.3** |
| ESOL | 1.387 | 1.195 | 1.386 | 1.187 | **1.151** |
| Lipo | 0.796 | 0.781 | 0.768 | 0.759 | **0.726** |
| Malaria | **1.105** | 1.116 | 1.143 | 1.145 | 1.110 |
| CEP | 1.341 | 1.384 | 1.367 | **1.334** | 1.338 |
| *Average* | 1.157 | 1.119 | 1.166 | 1.106 | **1.081** |

(b) GraphGPS Encoder (T)

| Dataset | SupLearn | AttrMask | MAM-A | MAM-VQ | MotifPred |
|---|---|---|---|---|---|
| Tox21 | 69.6 | 74.7 | 75.0 | 73.8 | **76.5** |
| ToxCast | 59.1 | 64.7 | 64.5 | 63.9 | **67.1** |
| Sider | 57.9 | 59.1 | **60.5** | 60.0 | 57.7 |
| MUV | 69.1 | 75.4 | 75.5 | 75.4 | **77.3** |
| HIV | 68.8 | 76.9 | 76.0 | 75.9 | **78.9** |
| BBBP | 59.8 | 68.1 | 67.6 | 65.8 | **68.2** |
| Bace | 70.3 | 81.2 | 79.9 | 80.5 | **84.3** |
| *Average* | 64.9 | 71.4 | 71.3 | 70.8 | **72.9** |
| ESOL | 1.036 | 1.194 | 1.356 | 1.297 | **0.984** |
| Lipo | 0.744 | 0.747 | 0.862 | 0.793 | **0.688** |
| Malaria | 1.130 | 1.105 | 1.123 | 1.115 | **1.084** |
| CEP | 1.689 | 1.260 | 1.408 | 1.337 | **1.139** |
| *Average* | 1.150 | 1.077 | 1.187 | 1.136 | **0.974** |

While node-level targets yields similar results, motif prediction consistently perform best, especially with GraphGPS. To explain this gap, we again utilize MI to study the statistical dependence between local and graph-level labels.

**Mutual Information Analysis**   Unlike Section 5.1, here MI is computed exactly by enumerating $(X, Y)$ over $\bigsqcup_G \mathcal{S}(G)$, which is equivalent to taking expectation under the uniform distribution of $X$.

For all three node-level labels, it is naturally guaranteed that the atoms in the downstream datasets come from the same element set as those in the pretraining data. However, for motif-level labels, such consistency is not inherently ensured, since downstream molecules may contain motifs that were not observed during pretraining. To ensure the validity of this analysis also applied for motif labels, we evaluated the coverage of downstream motifs within the pretraining motif vocabulary. On average, 67.3% of motif classes in each downstream dataset are found in the pretraining vocabulary. More importantly, across all datasets, over 92.9–99.5% of molecules contain at least 80% pretraining-seen motifs, demonstrating substantial overlap[5].

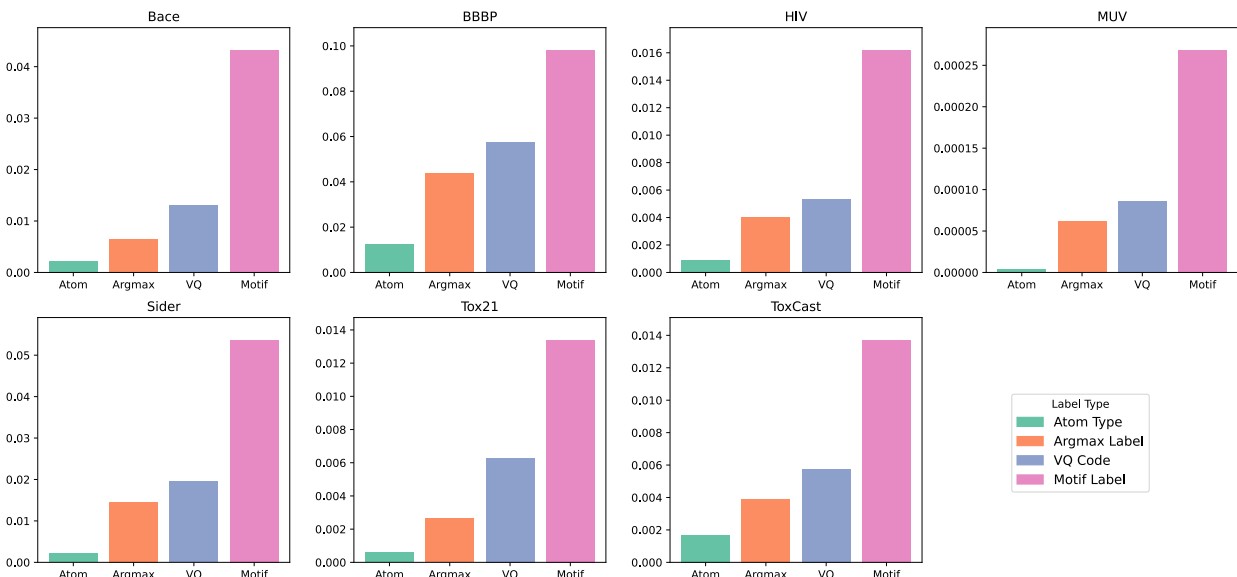

Figure 5: MI between local labels and graph label (per dataset)

---

[5]Per-dataset statistics are provided in Appendix A.9.

Figure 5 shows that motif labels yield consistently higher MI with graph-level labels than all node-level alternatives. Although absolute MI values are small due to the upper bound $H(Y)$, motif-level labels consistently explain 4–11% of the label entropy (see A.7).

**Analysis of Conditional Label Distributions** Although VQ and Argmax achieve higher average MI than atom types, this does not translate into proportional downstream gains (Table 7). To probe this discrepancy, we analyze conditional label distributions using Jensen–Shannon Divergence (Eq. 7), to test our hypothesis about the importance of low-frequency labels.

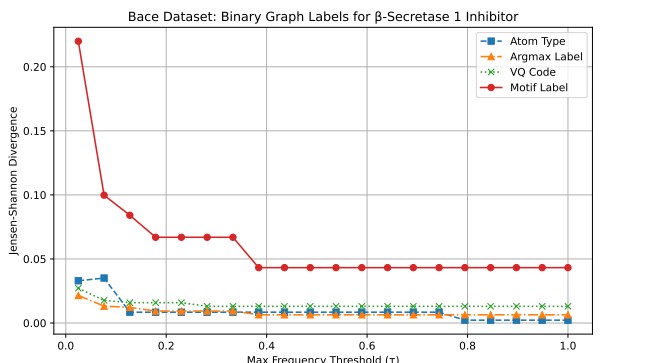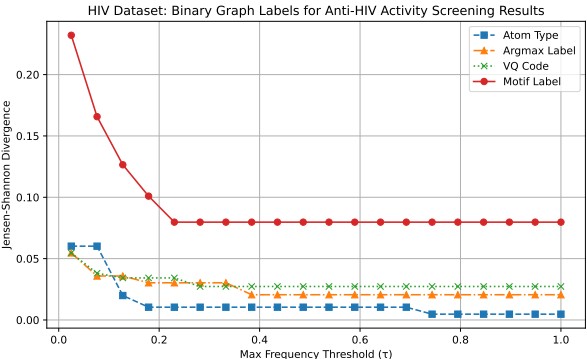

Figure 6: Jensen-Shannon Divergence (JSD) between conditional local label distributions $P(X|Y = 1, S_\tau)$ and $P(X|Y = 0, S_\tau)$ for two representative datasets: Bace and HIV.

As shown in Figure 6, we observe a consistent trend across both datasets: motif labels yield substantially higher JSD values than node-level labels when the threshold $\tau$ decreases, i.e., when we focus on increasingly rare local labels. Additional results for other labels from these datasets are provided in Appendix A.6.

This trend becomes especially pronounced at lower frequency thresholds (e.g., $\tau \leq 0.1$) where the divergence between $P(X|Y = 1, S_\tau)$ and $P(X|Y = 0, S_\tau)$ for motif labels sharply increases, whereas the JSD values for node-level labels remain relatively flat. We further include a shuffle-control experiment in Appendix A.6.2 to confirm that the observed gain is not attributable to vocabulary size. These results empirically support our hypothesis that low-frequency motifs carry more discriminative information with respect to molecular properties, likely because such motifs correspond to functional groups or structural patterns with specific bioactivity or chemical relevance.

In summary, motif labels provide more informative supervision than node-level alternatives. MI confirms stronger dependence on graph-level properties, and JSD highlights the discriminative power of rare motifs. These results support the intuition that functional substructures encode richer, task-relevant chemical semantics than individual atoms.

## 5.3 Encoder Architecture

This section investigates how the choice of encoder architecture influences the effectiveness of different pretraining targets. In addition to previously discussed objectives, we include MoAMa, which employs motif-level masking while predicting node-level targets. This mismatch in semantic granularity offers a valuable case study for understanding encoder-target alignment.

### 5.3.1 Performance Comparison Across Encoders

Figure 7 summarizes the downstream performance of key pretraining strategies when implemented with GIN versus GraphGPS encoders. A few observations emerge.

Firstly, both AttrMask and MotifPred show consistent gains when moving from GIN to GraphGPS. However, their implications differ. For AttrMask, GraphGPS reaches about 71.4% ROC-AUC, a level that can already

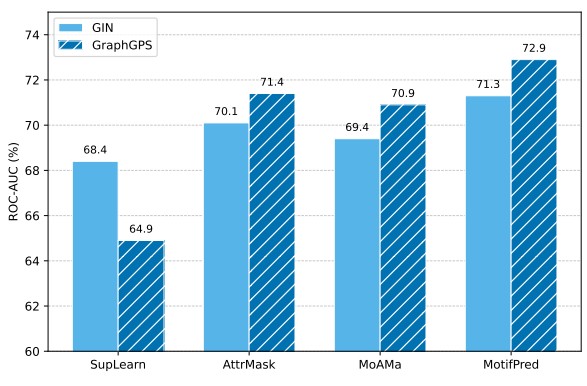

(a) Average ROC-AUC (↑) of Classification Tasks

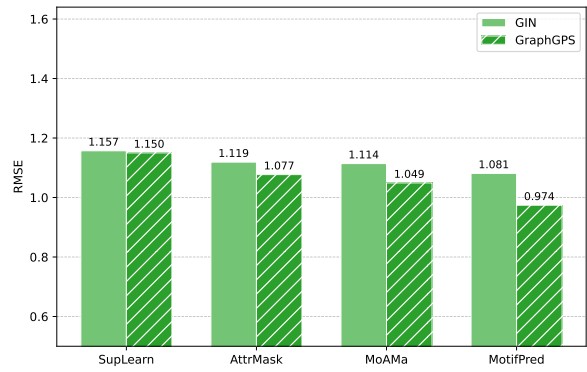

(b) Average RMSE (↓) of Regression Tasks

Figure 7: Comparison of downstream performance across encoder architectures (GIN vs. GraphGPS) for different masking designs.

be matched by GIN under alternative mask rates (see A.2). Thus, the observed improvement does not reflect a qualitatively new performance regime. In contrast, MotifPred with GraphGPS reliably achieves ∼72.9%, a higher regime that all examined methods cannot consistently reach with GIN.

Additionally, methods that use motif-level masking but retain atom-level prediction targets (i.e., MoAMa and MotifPred-A) show only marginal improvements with GraphGPS.

### 5.3.2 Encoder-Target Compatibility

These results suggest that structural complexity in masking is insufficient by itself; what matters more is the **semantic richness of the prediction target**. Even with motif-aware input perturbations, if the supervision remains atom-level, the task is fundamentally local and does not compel GraphGPS to exploit its global receptive field.

This comparison also echoes our earlier analysis of masking distributions. MoAMa, as described in Section 3.3, employs a 'non-adjacent' motif selection heuristic that increases pretraining time by about 220%. Despite this computational overhead, it provides no performance advantage over simple uniform sampling and can even perform slightly worse on GIN. The case of MoAMa reinforces our central conclusion: merely engineering a more complex masking distribution $P_{\mathcal{M}}$, without elevating the semantic richness of the prediction target $X$, is unlikely to be a promising direction for improving pre-trained graph models.

## 6 Discussion

### 6.1 A Formal Framework for Principled Comparison

A primary contribution of this work is the introduction of a formal probabilistic framework to deconstruct and analyze the design space of masking-based SSL on molecular graphs. By modeling the pretraining task as a process of fitting a random variable $X : \mathcal{S}(G) \to \mathcal{L}$ from structural units to a label space, we can move beyond ad-hoc comparisons and systematically investigate the distinct roles of its core components: the **masking distribution** ($P_{\mathcal{M}}$), the **prediction target** ($X$), and the **encoder architecture** ($f_\theta$). This principled formulation guides our entire investigation and enables us to isolate the impact of each design choice.

### 6.2 The Prediction Target Outweighs the Masking Distribution

Our systematic investigation reveals a clear hierarchy of importance among the design dimensions. The central finding of this work is that the choice of ***what* to predict (the prediction target) is substantially**

**more pivotal than the choice of *where* to mask (the masking distribution)**. Our probabilistic framework allows us to make the underlying hypothesis for sophisticated masking strategies explicit: that an optimal, non-uniform distribution $P_{\mathcal{M}}$ should make the pre-training signal $X$ more informative for the downstream task $Y$, which would manifest as a higher mutual information, $I(X; Y)$. However, our information-theoretic analysis of several representative heuristic and learnable strategies (Sec. 5.1.1) finds no evidence to support this hypothesis. This lack of a more informative signal, combined with their significant computational overhead (Appendix A.3), explains their failure to outperform simple uniform sampling in our experiments.

While this does not entirely preclude the existence of a more effective distribution, our work proposes a more resource-efficient methodology for future explorations. Rather than relying solely on expensive downstream evaluations, researchers can first leverage our framework as a low-cost litmus test: if a novel distribution demonstrably increases $I(X; Y)$, it warrants further investigation. Otherwise, our findings suggest that efforts are more fruitfully directed towards designing richer prediction targets.

### 6.3 The Critical Synergy Between Encoder and Target

A second key insight is the critical role of **synergy between the encoder architecture and the prediction target**. Our results consistently show that while standard MPNNs like GIN are well-suited for local, atom-level reconstruction tasks, their strong local inductive bias limits their ability to fully capitalize on semantically richer, non-local targets. In contrast, expressive Graph Transformer architectures, with their global attention mechanism, unlock significant performance gains when paired with motif-level prediction. This highlights that the benefits of a more powerful encoder are not universal but are contingent on being paired with a pretraining task that requires its advanced capabilities, such as modeling long-range dependencies to understand the semantics of a larger substructure.

### 6.4 Implications for Future Research: The Quest for Semantically Rich Targets

Our findings strongly advocate for shifting focus towards semantically richer prediction targets. This naturally raises the question: what constitutes semantic richness in the context of molecular SSL? Our work provides a comparative answer. While learned discrete tokens from methods like MAM represent a data-driven form of semantics, they appear less effective than human-curated chemical concepts like BRICS-defined motifs. Our information-theoretic analysis corroborates this, showing that motif labels have a stronger statistical dependence on downstream properties. This suggests that, at least for now, pretraining targets $X$ whose label space $\mathcal{L}$ is defined by **explicit, chemically-aware structural knowledge** provide a more potent supervisory signal than purely abstract, learned representations. An exciting avenue for future work could be the development of hybrid targets that combine the best of both worlds—learning to discover novel, meaningful substructures that go beyond traditional, human-defined motifs.

## 7 Conclusion

By formalizing the molecular graph masking pipeline within a probabilistic framework and leveraging information-theoretic measures to assess task alignment, we conducted a systematic investigation into the core design dimensions of self-supervised learning. Our investigation concludes that the various design dimensions do not hold equal weight: the choice of a semantically rich prediction target is the most critical driver of performance, whose full potential is only realized through a strong synergy with an expressive encoder architecture. In contrast, sophisticated masking distributions offer limited performance gains at a higher computational cost. These insights, derived from a principled and reproducible methodology, provide a clearer and more resource-efficient roadmap for developing the next generation of SSL methods for molecular property prediction.

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

# A Appendix

## A.1 Complete Results

This appendix contains the comprehensive results of our downstream evaluation. For clarity and easy reference, we first present Table 8, which summarizes the design choices for every method implemented in this study. The subsequent tables then provide the full, unabridged performance metrics (ROC-AUC for classification and RMSE for regression) for all variants of AttrMask across all MoleculeNet[6] benchmark tasks discussed in the main paper. Results for the variants of GraphMAE are presented in latter sections.

Table 8: Configurations for all implemented pre-training methods and their variants, categorized by their primary design choices. Checkmarks (✓) indicate the utilized components for each method.

| Method | Distribution | | | Prediction Target | | | | Loss | GNN Decoder | Re-mask |
|---|---|---|---|---|---|---|---|---|---|---|
| | Uni. | Heu. | Learn | Atom Type | Bond Type | Learned Token | Motif Label | | | |
| *Baselines based on AttrMask* | | | | | | | | | | |
| AttrMask | ✓ | | | ✓ | | | | CE | | |
| AttrMask-B | ✓ | | | ✓ | ✓ | | | CE | | |
| *Variants with Learned/Structured Targets* | | | | | | | | | | |
| MAM-A | ✓ | | | | | ✓ | | CE | | |
| MAM-VQ | ✓ | | | | | ✓ | | CE | | |
| MAM-A-B | ✓ | | | | ✓ | ✓ | | CE | | |
| MoAMa | | ✓ | | ✓ | | | | CE | | |
| MotifPred | ✓ | | | | | | ✓ | CE | | |
| MotifPred-P | | ✓ | | | | | ✓ | CE | | |
| MotifPred-L | | | ✓ | | | | ✓ | CE | | |
| MotifPred-A | | ✓ | | ✓ | | | | CE | | |
| *Auto-Encoding Variants (GraphMAE & StructMAE)* | | | | | | | | | | |
| GraphMAE | ✓ | | | ✓ | | | | SCE | ✓ | |
| GraphMAE-R | ✓ | | | ✓ | | | | SCE | ✓ | ✓ |
| GraphMAE-CE | ✓ | | | ✓ | | | | CE | ✓ | ✓ |
| StructMAE-P | | ✓ | | ✓ | | | | SCE | ✓ | |
| StructMAE-P-R | | ✓ | | ✓ | | | | SCE | ✓ | ✓ |
| StructMAE-P-CE | | ✓ | | ✓ | | | | CE | ✓ | |
| StructMAE-L | | | ✓ | ✓ | | | | SCE | ✓ | |
| StructMAE-L-R | | | ✓ | ✓ | | | | SCE | ✓ | ✓ |
| StructMAE-L-CE | | | ✓ | ✓ | | | | CE | ✓ | ✓ |

### A.1.1 MoleculeNet: Full Fine-tuning

| | Tox21 | ToxCast | Sider | MUV | HIV | BBBP | Bace | Average |
|---|---|---|---|---|---|---|---|---|
| # of data | 7831 | 8577 | 1427 | 93087 | 41127 | 2039 | 1513 | - |
| SupLearn | 73.9 (0.7) | 63.6 (0.6) | 57.7 (1.4) | 73.1 (1.7) | 74.3 (1.4) | 67.2 (2.5) | 68.8 (3.4) | 68.4 |
| SupLearn (T) | 69.6 (0.6) | 59.1 (1.3) | 57.9 (1.7) | 69.1 (0.9) | 68.8 (3.7) | 59.8 (3.4) | 70.3 (5.8) | 64.9 |
| AttrMask | 75.8 (0.5) | 64.3 (0.2) | 60.2 (1.1) | 72.3 (2.0) | 76.5 (1.6) | 63.4 (2.3) | 78.0 (1.0) | 70.1 |
| AttrMask-B | 76.1 (0.7) | 63.9 (0.5) | 59.3 (0.6) | 72.7 (1.5) | 77.6 (0.3) | 65.6 (1.8) | 77.1 (1.2) | 70.3 |
| MAM-A | 74.9 (0.8) | 61.7 (0.5) | 58.2 (0.3) | 77.8 (2.3) | 76.8 (1.8) | 65.4 (1.4) | 80.9 (1.1) | 70.8 |
| MAM-A-B | 76.0 (0.3) | 63.8 (0.3) | 59.3 (0.8) | 74.3 (2.0) | 76.5 (1.0) | 64.2 (2.7) | 77.8 (1.2) | 70.3 |
| MAM-VQ | 75.7 (0.4) | 63.3 (0.3) | 59.4 (0.7) | 76.0 (1.3) | 76.9 (1.1) | 64.6 (1.8) | 78.1 (1.2) | 70.6 |
| MotifPred-A | 76.3 (0.3) | 64.2 (0.4) | 57.6 (0.7) | 75.1 (1.3) | 76.9 (1.5) | 67.0 (0.9) | 80.1 (0.6) | 71.0 |
| MotifPred | 76.6 (0.6) | 64.5 (0.5) | 60.5 (0.7) | 76.8 (1.7) | 76.8 (0.4) | 64.7 (2.0) | 79.3 (5.0) | 71.3 |
| MotifPred-P | 75.6 (0.4) | 63.4 (0.4) | 59.2 (0.9) | 75.2 (3.1) | 77.3 (0.7) | 64.4 (2.1) | 76.1 (4.3) | 70.2 |
| MotifPred-L | 77.0 (0.3) | 64.4 (0.2) | 60.3 (0.7) | 77.6 (1.4) | 77.1 (1.5) | 64.1 (1.2) | 80.7 (1.6) | 71.6 |
| AttrMask (T) | 74.7 (0.4) | 64.7 (0.8) | 59.1 (0.9) | 75.4 (1.9) | 76.9 (1.3) | 68.1 (0.8) | 81.2 (2.5) | 71.4 |
| AttrMask-P (T) | 75.4 (1.0) | 65.6 (1.2) | 53.5 (1.3) | 76.8 (2.5) | 76.7 (2.1) | 70.4 (2.0) | 82.5 (1.3) | 71.6 |
| AttrMask-L (T) | 75.3 (1.1) | 64.9 (0.7) | 56.1 (0.9) | 77.3 (1.6) | 75.4 (1.7) | 67.6 (1.3) | 83.2 (0.5) | 71.4 |
| MAM-A (T) | 75.0 (1.1) | 64.5 (0.8) | 60.5 (1.2) | 75.5 (1.6) | 76.0 (1.3) | 67.6 (0.2) | 79.9 (2.4) | 71.3 |
| MAM-VQ (T) | 73.8 (1.0) | 63.9 (0.4) | 60.0 (1.6) | 75.4 (0.9) | 75.9 (1.8) | 65.8 (3.8) | 80.5 (2.0) | 70.8 |
| MotifPred-A (T) | 75.3 (1.1) | 64.9 (0.6) | 56.8 (2.0) | 74.6 (2.6) | 74.2 (1.2) | 69.7 (0.9) | 83.0 (1.3) | 71.2 |
| MotifPred (T) | 76.5 (0.4) | 67.1 (0.9) | 57.7 (1.6) | 77.3 (2.0) | 78.9 (1.1) | 68.2 (1.5) | 84.3 (2.2) | 72.9 |
| MotifPred-P (T) | 76.4 (0.6) | 66.4 (0.6) | 58.4 (1.6) | 76.4 (2.2) | 75.9 (0.1) | 64.7 (1.6) | 83.8 (2.0) | 71.7 |
| MotifPred-L (T) | 76.6 (0.8) | 65.4 (0.6) | 57.7 (0.6) | 78.1 (1.2) | 76.8 (0.7) | 69.8 (0.7) | 85.2 (1.1) | 72.8 |

Table 9: Comparison among AttrMask-based approaches with modified objectives (with standard deviation)

---
[6]We exclude the Clintox dataset from our evaluation due to its severe class imbalance and known data quality issues, which can lead to misleading performance metrics.

|  | ESOL | Lipophilicity | Malaria | CEP | Average RMSE |
|---|---|---|---|---|---|
| # of data | 1,128 | 4,200 | 9,999 | 29,978 | - |
| SupLearn | 1.387 (0.087) | 0.796 (0.019) | 1.105 (0.011) | 1.341 (0.010) | 1.157 |
| SupLearn (T) | 1.036 (0.084) | 0.744 (0.039) | 1.130 (0.007) | 1.689 (0.064) | 1.150 |
| AttrMask | 1.195 (0.024) | 0.781 (0.011) | 1.116 (0.002) | 1.384 (0.016) | 1.119 |
| AttrMask-B | 1.191 (0.028) | 0.759 (0.010) | 1.124 (0.020) | 1.343 (0.025) | 1.104 |
| MoAMa | 1.212 (0.022) | 0.773 (0.006) | 1.125 (0.009) | 1.344 (0.014) | 1.114 |
| MAM-A | 1.386 (0.020) | 0.768 (0.014) | 1.143 (0.023) | 1.367 (0.013) | 1.166 |
| MAM-A-B | 1.191 (0.038) | 0.759 (0.017) | 1.121 (0.012) | 1.340 (0.005) | 1.103 |
| MAM-VQ | 1.187 (0.025) | 0.759 (0.009) | 1.145 (0.008) | 1.334 (0.015) | 1.106 |
| MotifPred | 1.151 (0.027) | 0.726 (0.008) | 1.110 (0.009) | 1.338 (0.035) | 1.081 |
| AttrMask (T) | 1.194 (0.073) | 0.747 (0.015) | 1.105 (0.013) | 1.260 (0.027) | 1.077 |
| MoAMa (T) | 1.041 (0.051) | 0.777 (0.018) | 1.100 (0.013) | 1.279 (0.048) | 1.049 |
| MAM-A (T) | 1.356 (0.077) | 0.862 (0.034) | 1.123 (0.006) | 1.408 (0.028) | 1.187 |
| MAM-VQ (T) | 1.297 (0.029) | 0.793 (0.020) | 1.115 (0.010) | 1.337 (0.032) | 1.136 |
| MotifPred (T) | 0.984 (0.025) | 0.688 (0.007) | 1.084 (0.010) | 1.139 (0.010) | 0.974 |

Table 10: MoleculeNet: Regression Tasks (RMSE)

### A.1.2 Addtional Comparisons: Linear Probing

In the following tables, we freeze the pretrained parameters of GNN backbones, only the MLP classifier is trained during fine-tuning. The results are collected with 5 random seeds.

|  | Tox21 | ToxCast | Sider | MUV | HIV | BBBP | Bace | Average |
|---|---|---|---|---|---|---|---|---|
| # of data | 7831 | 8577 | 1427 | 93087 | 41127 | 2039 | 1513 | - |
| AttrMask | 70.5 (0.1) | 59.6 (0.1) | 52.6 (0.3) | 69.0 (0.7) | 66.6 (1.3) | 58.3 (0.1) | 61.3 (3.4) | 62.6 (0.6) |
| MotifPred | 69.1 (0.1) | 61.5 (0.2) | 50.5 (1.9) | **72.1** (0.9) | 64.7 (0.8) | 60.3 (0.3) | 65.5 (5.4) | 63.4 (0.7) |
| AttrMask (T) | **71.9** (0.1) | 60.3 (0.1) | 54.8 (0.0) | 64.2 (0.3) | 63.4 (0.8) | 53.8 (0.2) | 74.5 (0.4) | 63.3 (0.1) |
| MotifPred (T) | 71.8 (0.1) | **63.1** (0.1) | **59.0** (0.4) | 70.6 (0.2) | **74.4** (0.2) | **66.8** (0.1) | **77.4** (4.2) | **69.0** (0.6) |

Table 11: Linear Probing: among AttrMask-based approaches

|  | Tox21 | ToxCast | Sider | MUV | HIV | BBBP | Bace | Average |
|---|---|---|---|---|---|---|---|---|
| # of data | 7831 | 8577 | 1427 | 93087 | 41127 | 2039 | 1513 | - |
| GraphMAE | 68.5 (0.2) | **62.1** (0.3) | **58.5** (1.3) | 72.9 (1.9) | **66.3** (0.6) | **60.3** (2.9) | 73.0 (2.2) | **65.9** (0.7) |
| StructMAE-P | 67.9 (0.3) | 60.2 (0.3) | 56.0 (0.5) | 72.3 (1.6) | 64.6 (0.6) | 56.5 (0.4) | **76.3** (0.9) | 64.8 (0.2) |
| StructMAE-L | **70.6** (0.4) | 59.3 (0.4) | 56.4 (1.0) | **74.9** (2.0) | 61.9 (2.6) | 59.1 (0.4) | 64.2 (0.9) | 63.8 (0.4) |

Table 12: Linear Probing: among GraphMAE-based approaches

## A.2 Mask Ratio

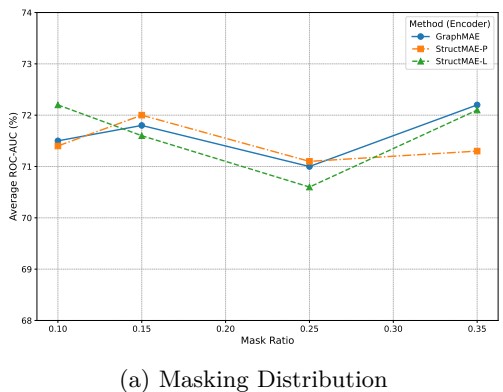

(a) Masking Distribution

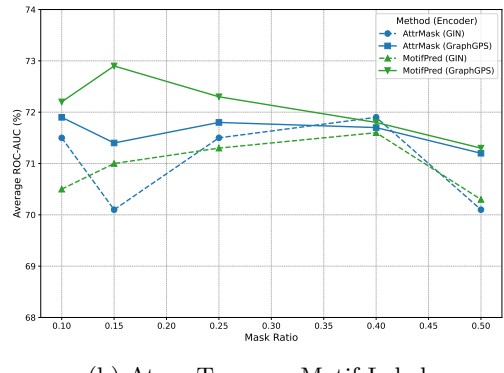

(b) Atom Type vs. Motif Label

Figure 8: Mask Ratio Sensitivity Analysis

The mask-ratio sensitivity analysis (Fig. 8) offers two key observations. First, although both atom-type prediction and motif prediction ultimately converge to comparable performance, their preferred mask ratios differ: MotifPred (T) achieves its peak at a relatively low ratio (0.15), while higher ratios (e.g., 0.40) lead to a noticeable drop. For MotifPred, a 0.40 ratio corresponds to masking 40% of atoms while recovering 80% of motifs. This more aggressive corruption likely disrupts the structural context needed for motif-level labels, while atom-level targets remain less sensitive.

Second, for GraphMAE and its variants, performance remains closely aligned across different masking distributions under all mask ratios. This consistency reinforces our earlier conclusion: modifying the sampling distribution provides little advantage over uniform masking, regardless of the corruption level.

## A.3 Pretraining Time Comparison

All models were pre-trained for 100 epochs on 2 million molecules from ZINC15 using a single NVIDIA A6000 GPU. For data loading, we utilized 8 parallel workers. The relative slowdown is calculated against the AttrMask (GIN) baseline.

Note that the training for GraphMAE is more efficient than AttrMask due to its pre-computation of one-hot vectors of atom attributes for training loss computation.

Table 13: Comparison of pretraining time for key model configurations.

| Method | Key Design Choice | Pretraining Time (Hours) | Relative Slowdown |
|---|---|---|---|
| AttrMask (GIN) | Uniform Masking, Atom Target (Baseline) | 11.7 | 1.0x |
| AttrMask (T) | Uniform Masking, Transformer Encoder | 18.3 | 1.6x |
| MoAMa (GIN) | Motif Masking (On-the-fly) | 37.2 | 3.2x |
| MoAMa (T) | Motif Masking, Transformer Encoder | 49.2 | 4.2x |
| GraphMAE (GIN) | Uniform Masking | 7.8 | 0.7x |
| StructMAE-P (GIN) | PageRank Masking (PageRank) | 24.4 | 2.1x |
| StructMAE-L (GIN) | Learnable Masking | 27.5 | 2.4x |
| MotifPred (GIN) | Motif Target (Pre-computed) | 18.6 | 1.6x |
| MotifPred (T) | Motif Target, Transformer Encoder | 23.5 | 2.0x |

### A.3.1 Asymptotic Cost of Masking Strategies

Let an input graph be $G = (V, E)$. The overall complexity of our pretraining pipeline is dominated by the encoder backbone, both GINE and GraphGPS scaling linearly with the graph size, i.e., $O(|V| + |E|)$ (Xu et al., 2019; Rampášek et al., 2022). Below we analyze the additional asymptotic cost introduced by different masking distributions.

- GraphMAE (Uniform). Random node indices are drawn uniformly and masked through vectorized assignment, adding at most $O(|V|)$ overhead. The total complexity remains $O(|V| + |E|)$.

- StructMAE-P (PageRank-based). Node importance scores are computed via power iteration for PageRank, $O(T|E|)$ with $T$ iterations to convergence. Two per-graph top-$k$ selections for masking incur $O(|V| \log |V|)$ each. The per-batch cost is therefore $O(T|E| + |V| \log |V|)$.

- StructMAE-L (Learnable). Replaces PageRank with an MLP + GNN scorer, $O(|V| + |E|)$, followed by the same two top-$k$ operations, giving $O(|V| \log |V| + |E|)$ overall.

All variants share the same memory footprint as the backbone, $O(|V| + |E|)$, since masking is performed online without pre-computation. In summary, non-uniform masking incurs additional $O(|V| \log |V|)$ cost for node scoring and sorting, explaining the observed computational overhead. See A.12 for details.

## A.4 Ablation Studies on Auxiliary Components

In this section, we conduct ablation studies on several auxiliary components proposed in the reproduced works to assess their impact on downstream performance.

| | Tox21 | ToxCast | Sider | MUV | HIV | BBBP | Bace | Average |
|---|---|---|---|---|---|---|---|---|
| # of data | 7831 | 8577 | 1427 | 93087 | 41127 | 2039 | 1513 | - |
| GraphMAE | 75.3 (0.6) | 64.1 (0.5) | 58.4 (0.5) | 74.9 (1.5) | 76.5 (1.6) | 67.2 (2.7) | 80.7 (2.5) | 71.0 |
| GraphMAE-R | 75.2 (0.4) | 63.9 (0.5) | 59.0 (0.9) | 74.2 (2.2) | 77.1 (1.3) | 64.1 (1.3) | 80.8 (1.4) | 70.6 |
| GraphMAE-CE | 76.3 (0.4) | 64.0 (0.4) | 58.2 (0.7) | 74.9 (3.4) | 76.0 (1.2) | 64.0 (1.9) | 82.1 (1.0) | 70.8 |
| StructMAE-P | 75.5 (0.6) | 63.6 (0.3) | 58.6 (0.8) | 73.7 (2.7) | 76.9 (0.9) | 67.6 (3.5) | 82.0 (1.1) | 71.1 |
| StructMAE-P-R | 75.6 (0.3) | 64.0 (0.2) | 59.2 (0.8) | 75.4 (1.0) | 76.5 (1.8) | 64.3 (1.9) | 81.7 (1.2) | 71.0 |
| StructMAE-P-CE | 75.1 (0.4) | 64.2 (0.4) | 59.9 (0.9) | 74.6 (2.0) | 76.3 (1.2) | 68.4 (2.0) | 83.2 (1.4) | 71.7 |
| StructMAE-L | 75.4 (0.6) | 63.9 (0.4) | 59.6 (0.9) | 73.6 (1.1) | 76.7 (1.5) | 65.2 (2.7) | 79.9 (0.7) | 70.6 |
| StructMAE-L-R | 75.2 (0.2) | 63.3 (0.5) | 59.6 (0.8) | 75.8 (1.1) | 76.0 (1.3) | 61.3 (2.0) | 78.1 (4.7) | 69.9 |
| StructMAE-L-CE | 76.3 (0.4) | 64.0 (0.4) | 58.2 (0.7) | 74.9 (3.4) | 76.0 (1.2) | 64.0 (1.9) | 82.1 (1.0) | 70.8 |

Table 14: Comparison among GraphMAE-based approaches

### A.4.1 Edge Masking

Our investigation also included edge attribute masking, where the model is pre-trained to predict the type of masked bonds. Similar to our findings on masking distributions, this strategy did not yield significant performance advantages (see Table 9 and 10). We attribute this to two primary factors. First, predicting a bond's type is a fundamentally local task. The graph topology and surrounding atoms often provide sufficient context for reconstruction. Second, the semantic information encoded in standard bond types (e.g., single, double) is inherently limited. Consequently, the supervisory signal generated from this task appears insufficient to drive the learning of powerful representations needed for complex graph-level properties.

### A.4.2 Decoder Architecture

The results show that replacing the MLP decoder with a GNN-based one brings no consistent difference under our current settings. This is likely because the pretraining task is relatively simple—predicting discrete atom-type labels rather than high-dimensional targets. More expressive decoders may become beneficial in more complex reconstruction settings, as suggested by SimSGT (Liu et al., 2023), though we leave such verification to future work.

### A.4.3 Loss Function

We compare the standard Cross-Entropy (CE) loss with the Scaled Cosine Error (SCE) loss, which was proposed by GraphMAE to down-weight easy examples. The comparison is made across three different masking distributions. As shown in Table 14, we consistently observe that models trained with the CE loss

(e.g., StructMAE-P-CE at 71.7%) outperform their counterparts trained with SCE (e.g., StructMAE-P-R at 71.0%). This suggests that for these atomic attribute reconstruction tasks, the standard CE loss remains a more effective and robust choice.

### A.4.4   Re-masking

The re-masking technique, introduced by GraphMAE, involves masking the latent representations of already-masked nodes before feeding them to the decoder. We evaluated this trick across the GraphMAE, StructMAE-P, and StructMAE-L frameworks. The results in Table 14 show no clear benefit from this technique. In all three pairs, the model with re-masking (denoted by the (-R) suffix) performs either comparably to or slightly worse than the model without it (e.g., GraphMAE: 71.0% vs. GraphMAE-R: 70.6%). We, therefore, conclude that the re-masking step, at least within our experimental setup, does not provide a consistent advantage and adds unnecessary complexity to the pretraining pipeline.

## A.5   Polaris Benchmarks

### A.5.1   A Case Study on a Low-Data Regime: The Polaris PKIS Benchmark

Our experiments on the Polaris PKIS (Elkins et al., 2016) benchmark largely reinforced the main conclusions from our broader study. As shown in Figure 9, we again found that sophisticated masking distributions (e.g., PageRank-based vs. Uniform) and variations among different node-level prediction targets (e.g., Atom Type vs. Argmax) offered no significant advantage over their simplest counterparts.

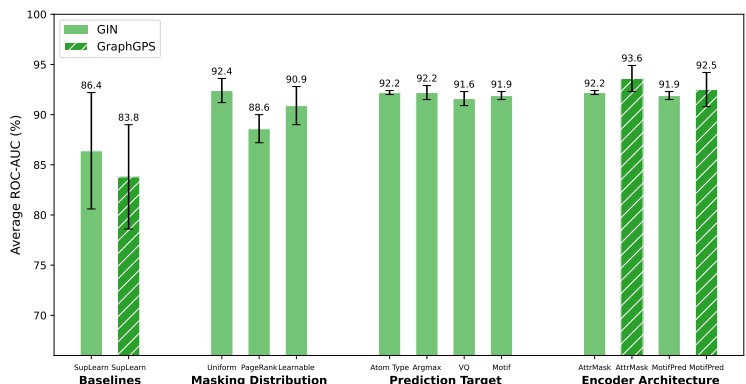

Figure 9: Average ROC-AUC (%) ± std on PKIS Classification over 5 Runs

However, this benchmark revealed one notable exception. In a direct reversal of the trend observed on larger datasets, the simpler AttrMask(T) model empirically outperformed the more powerful MotifPred(T). This phenomenon does not contradict our core findings. Instead, we attribute this performance inversion primarily to overfitting. Indeed, the PKIS dataset contains only 640 molecule. As illustrated in Figure 10, this is reflected in the training curves: MotifPred(T) converges significantly faster and to a near-perfect ROC-AUC on the training set, which indicates that its pre-trained features are indeed more powerfully aligned with the task.

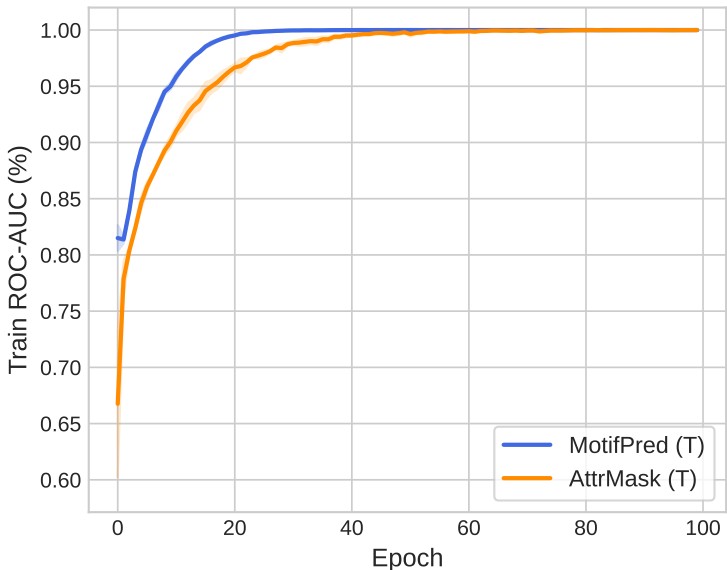

Figure 10: On PKIS, MotifPred(T) converges faster than AttrMask(T)

This result serves as a crucial case study highlighting that in data-scarce downstream applications, a model's robustness to overfitting can be a more decisive factor than the theoretical richness of its pretraining signal. The simpler AttrMask task may inadvertently act as a **regularizer**, leading to a less powerful but ultimately more generalizable model for this specific application.

### A.5.2 Additional Regression Results: The Polaris ADME Benchmark

| # of data | adme-microsomal 3,049 | adme-sol 2,173 | adme-ppb 115 | adme-perm 2,642 | Average - |
|---|---|---|---|---|---|
| SupLearn | 0.561 (0.018) | 0.58 (0.010) | 0.969 (0.174) | 0.668 (0.044) | 0.695 |
| SupLearn (T) | 0.534 (0.015) | 0.556 (0.017) | 0.771 (0.03) | 0.558 (0.021) | 0.605 |
| AttrMask | 0.536 (0.004) | 0.645 (0.021) | 0.706 (0.018) | 0.665 (0.016) | 0.638 |
| AttrMask-B | 0.534 (0.005) | 0.595 (0.010) | 0.593 (0.082) | 0.672 (0.014) | 0.599 |
| MAM-A | 0.567 (0.008) | 0.594 (0.010) | 0.622 (0.057) | 0.656 (0.021) | 0.610 |
| MAM-A-B | 0.545 (0.003) | 0.621 (0.012) | 0.521 (0.055) | 0.657 (0.026) | 0.586 |
| MoAMA | 0.579 (0.004) | 0.598 (0.007) | 0.570 (0.024) | 0.642 (0.009) | 0.597 |
| MotifPred | 0.582 (0.006) | 0.599 (0.008) | 0.621 (0.045) | 0.665 (0.006) | 0.617 |
| MotifPred-A | 0.558 (0.002) | 0.589 (0.010) | 0.672 (0.013) | 0.630 (0.007) | 0.612 |
| GraphMAE | 0.559 (0.014) | 0.608 (0.021) | 0.573 (0.078) | 0.631 (0.011) | 0.593 |
| GraphMAE-R | 0.557 (0.005) | 0.601 (0.012) | 0.567 (0.036) | 0.637 (0.006) | 0.591 |
| GraphMAE-CE | 0.562 (0.005) | 0.609 (0.011) | 0.626 (0.033) | 0.661 (0.009) | 0.615 |
| StructMAE-P | 0.559 (0.006) | 0.596 (0.004) | 0.546 (0.037) | 0.635 (0.010) | 0.584 |
| StructMAE-P-R | 0.587 (0.005) | 0.605 (0.008) | 0.599 (0.072) | 0.655 (0.010) | 0.611 |
| StructMAE-P-CE | 0.563 (0.011) | 0.599 (0.006) | 0.485 (0.027) | 0.651 (0.013) | 0.575 |
| StructMAE-L | 0.540 (0.009) | 0.598 (0.016) | 0.480 (0.033) | 0.638 (0.005) | 0.564 |
| StructMAE-L-R | 0.544 (0.004) | 0.610 (0.010) | 0.534 (0.093) | 0.631 (0.015) | 0.580 |
| StructMAE-L-CE | 0.541 (0.003) | 0.592 (0.011) | 0.610 (0.075) | 0.645 (0.005) | 0.597 |
| AttrMask (T) | 0.545 (0.015) | 0.588 (0.014) | 0.601 (0.044) | 0.596 (0.006) | 0.583 |
| MotifPred (T) | 0.506 (0.009) | 0.592 (0.017) | 0.615 (0.017) | 0.542 (0.012) | **0.564** |

Table 15: Results on ADME (RMSE)

## A.6    Additional Analysis of Conditional Label Distributions

### A.6.1    JSD Results across All Classification Datasets

Figure 11: Jensen-Shannon Divergence (JSD) between conditional local label distributions $P(X|Y = 1, S_\tau)$ and $P(X|Y = 0, S_\tau)$ for all evaluated classification datasets. The x-axis represents the maximum frequency threshold ($\tau$) for including labels in the analysis, and the y-axis represents the JSD value.

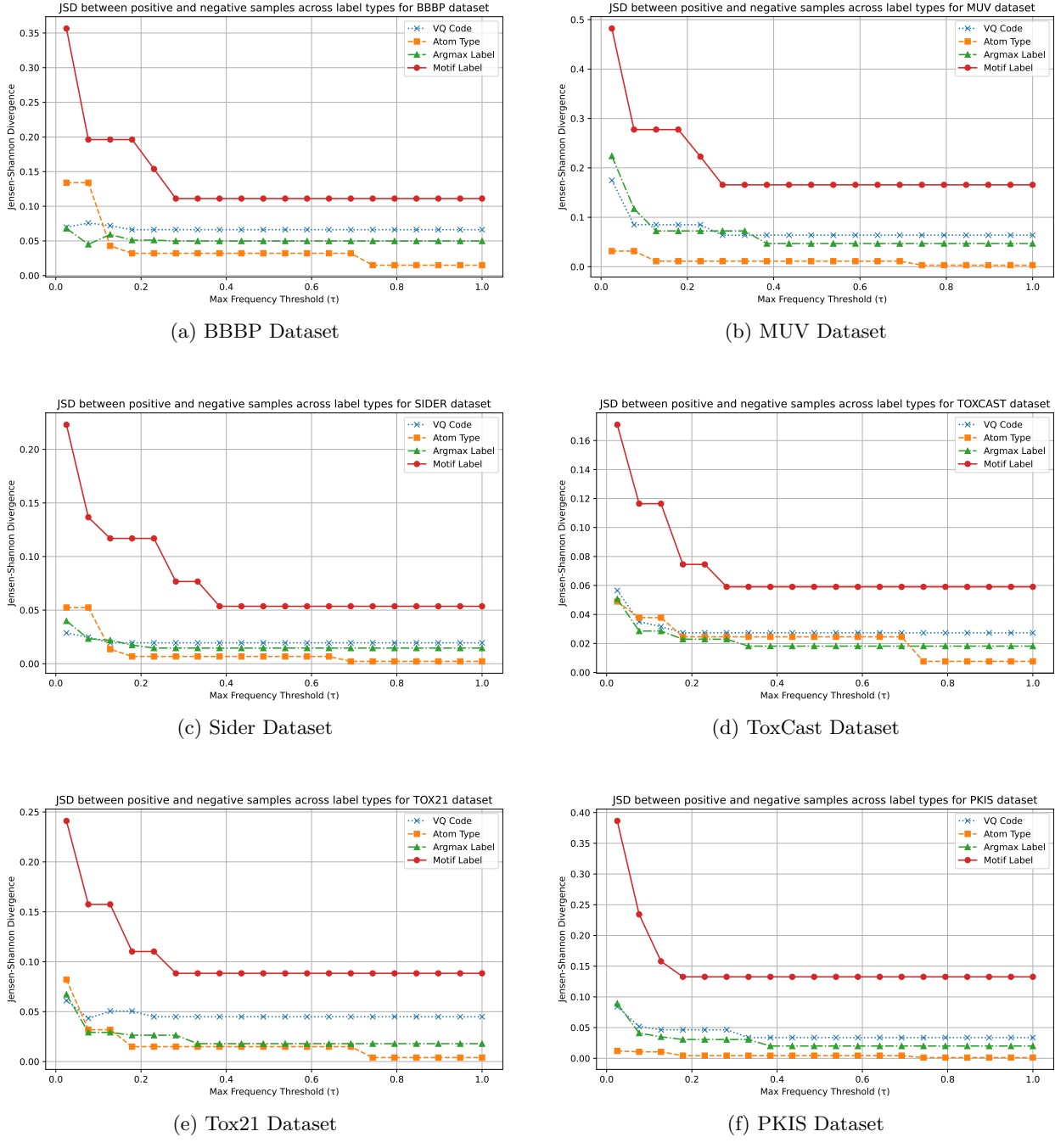

(a) BBBP Dataset

(b) MUV Dataset

(c) Sider Dataset

(d) ToxCast Dataset

(e) Tox21 Dataset

(f) PKIS Dataset

### A.6.2 Sanity Check with Shuffled Labels

To test whether the higher MI/JSD for motifs could be explained by the number of classes rather than semantic informativeness, we performed a sanity check by randomly permuting the motif labels across molecules, while keeping the graph labels $Y$ (positive vs. negative) fixed.

As shown in Table 16 and Figure 12, motif MI/JSD collapses to the atom-level baseline. This confirms that the observed effect reflects semantic informativeness rather than mere class-cardinality.

Table 16: MI across prediction targets with shuffled motif MI (5 random seeds)

| Dataset | Motif (orig.) | Motif (shuf.) | Atom type | Argmax | VQ |
|---------|--------------|---------------|-----------|--------|-----|
| Bace | **0.0433** | 0.0157±0.0004 | 0.0022 | 0.0064 | 0.0130 |
| BBBP | **0.0982** | 0.0250±0.0009 | 0.0127 | 0.0437 | 0.0575 |
| HIV | **0.0162** | 0.0057±0.0001 | 0.0009 | 0.0040 | 0.0053 |
| MUV | **0.0003** | 0.0002±0.0000 | 0.0000 | 0.0001 | 0.0001 |
| PKIS | **0.0583** | 0.0287±0.0022 | 0.0005 | 0.0086 | 0.0141 |
| Tox21 | **0.0134** | 0.0059±0.0005 | 0.0006 | 0.0027 | 0.0063 |
| ToxCast | **0.0137** | 0.0067±0.0004 | 0.0017 | 0.0039 | 0.0057 |

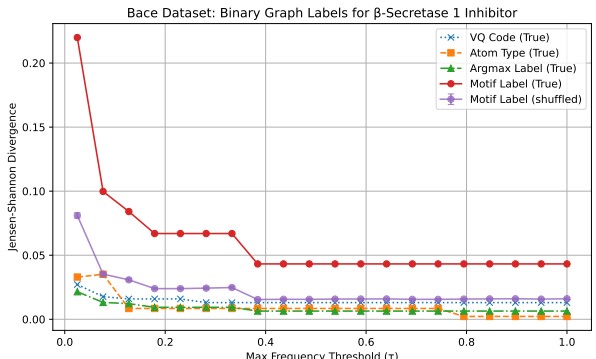 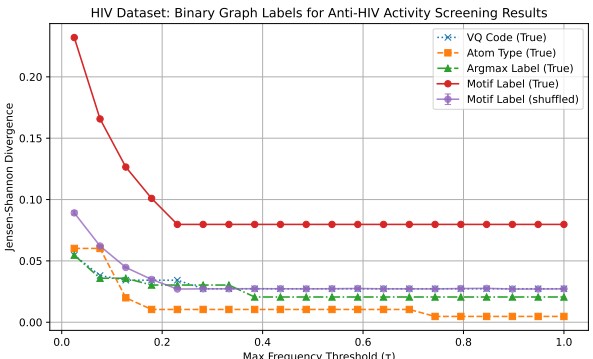

Figure 12: JSD curves with shuffled motif labels

### A.7 Mutual Information Statistics

By definition, $I(X;Y) \leq H(Y)$. Here we summarize the $H(Y)$ in the downstream datasets. This indicates the relative informativeness of motif labels is robust despite small absolute MI values.

Table 17: Upper bounds and relative gain of $I(X_{\text{motif}}, Y)$

| Dataset | $H(Y)$ | $I(X_{\text{motif}}; Y)$ | Relative Gain |
|---------|--------|--------------------------|---------------|
| Bace | 0.999 | 0.043 | 4.3% |
| BBBP | 0.918 | 0.098 | 10.7% |
| HIV | 0.279 | 0.016 | 5.7% |
| MUV | 0.0033 | 0.0003 | 9.1% |
| Tox21 | 0.210 | 0.013 | 5.7% |
| Toxcast | 0.316 | 0.014 | 4.3% |
| Sider | 0.999 | 0.054 | 5.4% |

### A.8 More Configuration

**Decoder Output Dimensions** In our implementation, decoder modules are configured to predict masked node, edge or motif labels. For each prediction task, the decoder's output dimension is set to match the number of possible categorical labels in the dataset. The following table summarizes the label dimensions for node-, edge- and motif-level reconstruction:

Table 18: Decoder output dimensions used for different level of prediction tasks.

| Prediction Target | Label Type | Output Dimension |
|---|---|---|
| Node attribute | Atom type | 119 |
| Edge attribute | Bond type | 4 |
| Motif attribute | Motif label | 35,082 |

### A.9 Motif Statistics

Here we list the percentage of overlapping motif in the **vocabularies** of the pretraining set and the downstream sets used for analysis in Section 5.2.

Table 19: Statistics of Motif Vocabularies

|  | Tox21 | ToxCast | Sider | MUV | HIV | BBBP | Bace | PKIS | ZINC |
|---|---|---|---|---|---|---|---|---|---|
| Vocab size | 1,219 | 1,284 | 737 | 4,721 | 6,794 | 738 | 389 | 266 | 35,082 |
| Intersection size | 943 | 964 | 465 | 3,914 | 2,994 | 515 | 220 | 185 | - |
| Overlap ratio (%) | 77.4 | 75.1 | 63.1 | 82.9 | 44.1 | 69.8 | 56.6 | 69.6 | - |

The overlap ratio is computed as the percentage of motifs in each downstream dataset that also appear in the pretraining vocabulary (i.e., $\frac{|\mathcal{V}_{\text{down}} \cap \mathcal{V}_{\text{pretrain}}|}{|\mathcal{V}_{\text{down}}|}$). This reflects how well the pretrained motif space covers the downstream distributions.

To further understand how many motifs in the downstream datasets have been seen in the pretraining, we compute the coverage ratio

$$r(G) = \frac{\#\{\text{motifs in } G \text{ seen in pretraining}\}}{\#\{\text{motifs in } G\}}$$

Across all downstream datasets, over 92.9–99.5% of molecules satisfy $r(G) \geq 0.8$ (i.e., at least 80% of their motifs were seen during pretraining). Conversely, true cold-start cases are extremely rare: only 0–0.7% of molecules in most datasets (and at most 2.5% in one dataset) have $r(G) \leq 0.2$.

Table 20: Per-molecule coverage statistics of $r(G)$ across downstream datasets.

| Dataset | Median | IQR | Mean | Std $r$ | % $r \geq 0.8$ | % $r \leq 0.2$ |
|---|---|---|---|---|---|---|
| sider | 1.0000 | 0.0000 | 0.9604 | 0.1648 | 95.30 | 2.52 |
| toxcast | 1.0000 | 0.0000 | 0.9764 | 0.1056 | 96.57 | 0.69 |
| tox21 | 1.0000 | 0.0000 | 0.9791 | 0.1015 | 97.02 | 0.68 |
| hiv | 1.0000 | 0.0000 | 0.9648 | 0.1033 | 92.89 | 0.20 |
| bbbp | 1.0000 | 0.0000 | 0.9868 | 0.0559 | 97.06 | 0.00 |
| bace | 1.0000 | 0.1250 | 0.9542 | 0.0748 | 98.74 | 0.00 |
| muv | 1.0000 | 0.0000 | 0.9962 | 0.0284 | 99.45 | 0.00 |
| pkis | 1.0000 | 0.0000 | 0.9562 | 0.0793 | 97.66 | 0.00 |

## A.10 Additional Implementation Details for Masking Distributions

We provide additional details on how heuristic (PageRank-based) and learnable masking distributions were adapted to different pretraining settings.

For the heuristic setting, starting from the original PageRank-based masking in StructMAE-P (see Algorithm 1), we implemented analogous variants for AttrMask and MotifPred. In the case of MotifPred, PageRank scores were computed on a coarsened version of the molecular graph, where each node represents a motif.

For the learnable setting, recall that StructMAE-L trains its masking scorer by propagating gradients through the encoder's unmasked node embeddings. This strategy, however, is not directly applicable to architectures with a simple linear decoder that only consumes masked node embeddings. We therefore adapted the learning mechanism in two cases:

- **AttrMask-L (T):** With a linear decoder, gradient flow is enabled by attaching the learned scores directly to the **masked** node embeddings from the encoder.

- **MotifPred-L (T):** A motif-level mask scorer is trained via a minimax-style objective: it maximizes the prediction loss while keeping the encoder frozen. To ensure differentiability, Gumbel-softmax-based scores are attached to the masked motif embeddings before loss computation.

These variants ensure that gradients can be propagated properly under different architectures. They are provided solely for completeness and reproducibility, and are not intended as novel contributions of this work.

## A.11 Implementation Details of MAM-VQ

As mentioned in Section 3.3.2, the label generation for MAM-VQ involves a nuanced two-stage process centered on its vector quantization (VQ) codebook.

**Stage 1: Tokenizer and Codebook Pretraining.** The VQ codebook and its corresponding GNN tokenizer are first jointly pretrained using a *group VQ* strategy. In this stage, the codebook is partitioned into four sub-codebooks based on atom type: one each for Carbon, Nitrogen, and Oxygen, and a fourth for all other elements. The search for the nearest codebook vector is constrained to the relevant partition (as illustrated in Figure 2).

**Stage 2: Main Encoder Pretraining.** However, for the main pretraining of the GNN encoder—the stage evaluated in our study—a different approach is taken. The pretrained VQ codebook is frozen, and the group constraint is **removed**. The prediction target for a masked atom is determined by a global search for the nearest vector across the entire codebook. While the original paper does not elaborate on this design choice, it is presumably intended to create a more challenging and effective reconstruction task for the main encoder. Our evaluation faithfully implements this two-stage procedure to ensure a fair comparison.

## A.12 Detailed Procedures for StructMAE Variants

For completeness, we include the full pseudocode for the two StructMAE variants described in Section 3.3. Both variants rely on a *perturbed top-k selection* scheme: after computing importance scores for nodes, the top-$k$ candidates are perturbed with random noise, and the final masked set is chosen according to the adjusted scores. The perturbation strength $\beta$ and annealed mask rate $\gamma_i$ follow the original settings.

---

**Algorithm 1:** Perturbed PageRank-Based Masking (StructMAE-P)

---

**Input:** Graph $G = (V, E)$; mask rate $\gamma$; epoch $i$ (max $E$); perturbation $\beta$
**Output:** Mask nodes $V_M \subset V$

**1** Compute PageRank scores $p[v]$ for all $v \in V$;
**2** Anneal effective mask rate: $\gamma_i \leftarrow \gamma \cdot \sqrt{i/E}$;
**3** Let $\mathcal{C}_i \leftarrow \texttt{TopK}_{v \in V}(p[v], \gamma_i)$;
**4** Sample random noise $s[v] \sim \mathcal{U}(0, 1)$ for all $v$;
**5** Add perturbation $s[v] \leftarrow s[v] + \beta$ for $v \in \mathcal{C}_i$;
**6** Select final nodes $V_M \leftarrow \texttt{TopK}_{v \in V}(s[v], \gamma)$;
**7 return** $V_M$

---

**Algorithm 2:** Perturbed Learnable Masking (StructMAE-L)

---

**Input:** Graph $G = (V, E)$; GNN/MLP scorers `gnn_scr`, `mlp_scr`; mask rate $\gamma$; epoch $i$
**Output:** Mask nodes $V_M \subset V$, scores $s \in \mathbb{R}^{|V|}$

**1** Compute learned scores $l[v] = \texttt{gnn\_scr}(v) + \lambda \cdot \texttt{mlp\_scr}(v)$;
**2** Anneal effective mask rate: $\gamma_i \leftarrow \gamma \cdot \sqrt{i/E}$;
**3** Let $\mathcal{C}_i \leftarrow \texttt{TopK}_{v \in V}(l[v], \gamma_i)$;
**4** Sample random noise $s[v] \sim \mathcal{U}(0, 1)$ for all $v$;
**5** Add perturbation $s[v] \leftarrow s[v] + \beta$ for $v \in \mathcal{C}_i$;
**6** Select final nodes $V_M \leftarrow \texttt{TopK}_{v \in V}(s[v], \gamma)$;
**7 return** $V_M$, $s$

---

In our implementation, $\beta = 0.25$ for StructMAE-P and $\beta = 0.5$, $\lambda = 1$ for StructMAE-L. The differentiability of StructMAE-L relies on passing gradients through the unmasked node embeddings, which are multiplied by the learned scores before being decoded by the GNN.

### A.13 Implementation Adaptations for Baselines

Table 21: Adaptations applied to baseline methods under our unified framework.

| Method | Adaptations |
|---|---|
| MAM-A/VQ (MoleBERT) | Remove TMCL loss; created only one masked view per batch |
| StructMAE-P/L | Changed default mask rate from 0.5 to 0.25 (aligned with GraphMAE) |
| MoAMa | Omitted auxiliary loss based on Tanimoto similarity |
| MotifPred (ReaCTMask) | Pretrained only on single molecules; adapted to GIN encoder |

