# OpenReview forum: "Self-Supervised Learning on Molecular Graphs: A Systematic Investigation of Masking Design"
_TMLR — Accepted by TMLR_

### Review · Reviewer_s6hh · 2025-08-31

**Summary Of Contributions:**

This paper systematically investigates masking-based self-supervised learning for molecular graphs. It shows that masking distributions bring little benefit over uniform masking, while prediction targets—especially motif-level labels—are far more influential, especially combined with Graph Transformer encoders. Beyond performance, the use of an information-theoretic analysis to interpret core design is particularly impressive.

**Weaknesses**

The interpretations of the results tend to be somewhat overstated and at times rely on unsupported inferences, which weakens the strength of the conclusions.

- The relative MI comparison suggests motifs are more informative (Table 5), but the absolute MI values are quite small. It is questionable whether such differences are practically meaningful without stronger statistical/other evidence.
- With about 40% of motifs unseen (64.3% overlap), this is not a negligible gap. In molecular tasks, such cold-start cases are critical and could significantly affect performance.
- Figure 6 shows higher JSD for motifs, but this may partly result from their larger structural size. The authors should clarify whether the effect reflects true semantic informativeness or simply size.
- In Figure 7, the performance gains from GIN to GraphGPS are quite modest—1.3 for AttrMask, 1.5 for MoAMa, and 1.6 for MotifPred. These differences are small, yet the paper presents them as “notable improvements,” which seems overstated.
- While the emphasis is on human-curated motifs, the comparison was at the atom level. Further analysis is needed to determine whether this is due to size or whether it is clearly based on external knowledge.

Various sections are insufficiently elaborated, with important details either missing or not clearly presented.

- Overall, the categorization/mapping between the core design and various models should be clear, but they are listed in a disorganized manner. A summary or categorization table organizing the models along the key dimensions—masking distribution (uniform, heuristic, learnable), prediction target (node or motif), and encoder architecture (GIN, Transformer)—would make the experimental design much clearer. (GraphMAE, StructMAE-P, StructMAE-L, MotifPred, MotifPred (T), AttrMask (T), AttrMask-L (T), AttrMask-P (T), MotifPred-L (T), SupLearn(?), …)
- The abbreviations used in the manuscript (e.g., Table 6 Sup, AM, MA, VQ, MP, and (T)) are not clearly explained in the text.
- Overall, the terminology in the paper is not well unified and feels inconsistent, which makes the presentation confusing. (Table 6: AM, MA, VQ, MP, in Figure 5: Atom, Argmax, VQ, Motif), and the order of factors is also disorganized (e.g., legend order in Figure 5, Figure 6)


**Strengths**

The attempt of an information-theoretic analysis (MI and JSD) to quantify the alignment between pretraining and downstream finetuning is impressive.

**Audience:**

Yes

**Audience Explanation:**

The work systematically investigates core design choices in masking-based self-supervised learning for molecular graphs. Some individuals in TMLR’s audience would be interested in this paper

**Broader Impact Concerns:**

This work does not raise direct ethical concerns, as it primarily presents methodological advances in self-supervised learning for molecular graphs.

**Claims And Evidence:**

Yes

**Claims Explanation:**

This is a well-designed and carefully implemented study that provides valuable insights into the relative importance of masking design choices. However, the paper sometimes overstates its conclusions, and important details are either missing or insufficiently explained. Addressing these issues would improve both clarity and credibility.

**Requested Changes:**

The list of proposed adjustments to the submission is critical to enhance the quality of the manuscript and to recommend for acceptance.
Authors need to address the weaknesses and ambiguities mentioned above.

---

> ### Author Response · Authors · 2025-10-04
> **Author Response to Reviewer s6hh**
>
> We thank the reviewer for their thoughtful and constructive feedback. Below, we provide our responses to each comment. All addtional experiments are conducted with 5 random seeds. A summary table of the corresponding manuscript revisions is included at the end.
>
> ## 1. Absolute MI values are quite small
>
> > The relative MI comparison suggests motifs are more informative (Table 5), but the absolute MI values are quite small.
> >
>
> The absolute MI values are small, since $I(X;Y)\leq H(Y)$ and the label entropy $H(Y)$ itself is small. We report the estimated $H(Y)$ in each dataset.
>
> As shown below, motif-level targets consistently explain 4–11% of the label entropy across datasets. This demonstrates that, although the absolute MI values are small due to the entropy bound, the relative gain is statistically meaningful and aligns with downstream improvements.
>
> |  | Bace | BBBP | HIV | MUV | Tox21 | Toxcast | Sider |
> | --- | --- | --- | --- | --- | --- | --- | --- |
> | H(Y) | 0.999 | 0.918 | 0.279 | 0.0033 | 0.210 | 0.316 | 0.999 |
> | I(Motif;Y) | 0.043 | 0.098 | 0.016 | 0.0003 | 0.013 | 0.014 | 0.054 |
> | Relative Gain | 4.3% | 10.7% | 5.7% | 9.1% | 5.7% | 4.3% | 5.4% |
>
> ## 2. Cold-start coverage at the molecule level.
>
> > about 40% of motifs unseen ... not a negligible gap.
> >
>
> While type-level motif overlap with the pretraining corpus is about 64.3%, what matters for downstream prediction is **per-molecule coverage**. We therefore computed the coverage ratio
>
> $r(G) = \frac{\textrm{motifs in G seen in pretraining}}{\textrm{motifs in G}}$
>
> Across all downstream datasets, over 92.9–99.5% of molecules satisfy $r(G) \geq 0.8$ (i.e., at least 80% of their motifs were seen during pretraining). Conversely, true cold-start cases are extremely rare: only 0–0.7% of molecules in most datasets (and at most 2.5% in one dataset) have $r(G) \leq 0.2$). Thus, despite ~40% of motif *types* being unseen globally, **the vast majority of test molecules are not in a true cold-start regime**; only a small tail of molecules has low coverage.
>
> We added the full table of per-molecule coverage statistics to A.9 to clarify this point and avoid any misinterpretation.
>
> ## 3. High JSD is not simply due to size
>
> > Figure 6 shows higher JSD for motifs, but this may partly result from their larger structural size.
> >
>
> Our emphasis on motifs is precisely because they are human-curated substructures (functional groups/common building blocks) intended to encode external domain knowledge beyond atom labels. We conducted the following analysis to disentangle the cause of the improvements.
>
> We added Figure 12 in A.6.2, when we shuffle motif labels between positive and negative molecules, the JSD curve collapses to the same level as atom-level labels. Moreover, note that in the Bace dataset the total number of motif classes (~400) is actually **smaller** than the 512 atom-level codes used in MAM-A/VQ, yet motifs still achieve higher JSD. These results confirm that the observed effect reflects semantic informativeness rather than structural size.
>
> ## 4. Wording about improvements
>
> > In Figure 7, the performance gains from GIN to GraphGPS are quite modest—1.3 for AttrMask, 1.5 for MoAMa, and 1.6 for MotifPred. These differences are small, yet the paper presents them as “notable improvements,” which seems overstated.
> >
>
> We agree that phrases like *“notable improvements”* may overstate the magnitude. We have revised the wording of relevant sentences in Section 5.3. Our intent was not to exaggerate the absolute margins, but to emphasize that MotifPred consistently yields stronger representations compared to atom-level baselines.
>
> This becomes clearer in a frozen-backbone evaluation (encoder fixed, only a classifier trained). We have added these results to the appendix to clarify that the improvements are robust and not artifacts of fine-tuning.
>
> | Method (frozen backbone) | GraphGPS (ROC-AUC) | GIN (ROC-AUC) |
> | --- | --- | --- |
> | AttrMask | 63.3 (0.1) | 62.6 (0.6) |
> | MotifPred | 69.0 (0.6) | 63.4 (0.7) |
>
> ## 5. Presentation
>
> > Various sections are insufficiently elaborated…SupLearn(?)… and the order of factors is also disorganized (e.g., legend order in Figure 5, Figure 6)
> >
>
> We acknowledge that some parts of the manuscript could be clearer. The *SupLearn* refers to the random initialized baseline without pretraining. In the revision, we have:
>
> (i) Added a categorization table in Sec. 4 on the implemented methods
>
> (ii) Aligned the legends in Figure 5 and 6.
>
> > Table 6: AM, MA, VQ, MP, in Figure 5: Atom, Argmax, VQ, Motif
> >
>
> We note that Figure 5 analyzes the MI of graph label $Y$ with pretraining *target type* $X$ (Atom, Argmax, VQ, Motif), whereas Table 6 lists the results of *pretraining methods* that were supervised on these targets. We have updated Table 7 to replace abbreviation with full method names.
>
> Due to length constraints, we provide the revision table in a separate comment below.

---

> ### Author Response · Authors · 2025-10-04
> **Revision Table**
>
> ## Summary of Revision
>
> | Section  | Modification | Purpose |
> | --- | --- | --- |
> | App. A.7 | Added note $I(X;Y)\leq H(Y)$, report relative gains over the upper bound | Clarify MI bound, extra evidence that motif MI is relatively significant |
> | Sec 5.2.1 & App. A.9 | Added statistics on coverage of pretrained motif classes | Clarify most motifs in the downstream dataset have been seen during pretraining |
> | App. A.6.2 | Shuffled-label sanity check | Show motif MI/JSD advantage is semantic, not size |
> | Sec. 5.3.1 & App. A.1.2 | Revise wording and added frozen-backbone results | Clarify the improvement is to reach a higher regime and the absolute improvement is more obvious when looking at the model weight right after pretraining |
> | Sec. 4  | added a categorization table with key elements in examined methods | For clarity and conciseness |
> | Sec 5.2.1 | Removed abbreviations and resize table; aligned legend of Figure 5 and 6 | For clarity  |

---

### Review · Reviewer_rV3K · 2025-09-01

**Summary Of Contributions:**

The paper provides a comprehensive analysis of various masking strategies in self-supervised learning for molecular representation learning. It formalizes the pretraining process in a probabilistic framework and evaluates the impact of different masking distributions, prediction targets, and encoder architectures on downstream molecular property prediction tasks.

Pros:
1. The paper introduces a unified probabilistic framework to systematically analyze and compare different masking strategies. 2. It conducts a thorough evaluation across multiple dimensions (masking distribution, prediction target, encoder architecture) under controlled settings. 3. The findings offer practical guidance for developing more effective SSL methods, particularly highlighting the importance of prediction targets and encoder architecture.

Cons:
1. The study shows that complex masking strategies significantly increase computational costs. Analysis of time and space complexity is necessary to better understand the trade-offs. 2. The hypothesis that non-uniform masking distributions would outperform uniform sampling wasn't supported by the results. A deeper analysis of why these strategies fail to deliver benefits could be insightful. 3. Edge attribute masking did not provide significant performance advantages. Under what conditions or types of tasks might edge masking be more beneficial? 4. The article mainly discusses masking strategy. It is recommended to introduce related methods: MOAT: Graph Prompting for 3D Molecular Graphs; SMI-Editor: Edit-based SMILES Language Model with Fragment-level Supervision. 5. The impact of using a more expressive GNN-based decoder versus a simple MLP decoder was secondary to hyperparameter tuning. Is a more detailed comparison of different decoder architectures needed to clarify their respective roles?

**Audience:**

Yes

**Audience Explanation:**

The paper provides a comprehensive analysis of various masking strategies in self-supervised learning for molecular representation learning. It formalizes the pretraining process in a probabilistic framework and evaluates the impact of different masking distributions, prediction targets, and encoder architectures on downstream molecular property prediction tasks.

**Claims And Evidence:**

Yes

**Claims Explanation:**

See above.

**Requested Changes:**

1. The study shows that complex masking strategies significantly increase computational costs. Analysis of time and space complexity is necessary to better understand the trade-offs. 2. The hypothesis that non-uniform masking distributions would outperform uniform sampling wasn't supported by the results. A deeper analysis of why these strategies fail to deliver benefits could be insightful. 3. Edge attribute masking did not provide significant performance advantages. Under what conditions or types of tasks might edge masking be more beneficial? 4. The article mainly discusses masking strategy. It is recommended to introduce related methods: MOAT: Graph Prompting for 3D Molecular Graphs; SMI-Editor: Edit-based SMILES Language Model with Fragment-level Supervision. 5. The impact of using a more expressive GNN-based decoder versus a simple MLP decoder was secondary to hyperparameter tuning. Is a more detailed comparison of different decoder architectures needed to clarify their respective roles?

---

> ### Author Response · Authors · 2025-10-04
>
> We thank the reviewer for their thoughtful feedback. Below, we provide our responses to each comment. A summary table of the corresponding manuscript revisions is included at the end.
>
> ## 1. Asymptotic analysis
>
> > …complex masking strategies significantly increase computational costs. Analysis of time and space complexity is necessary...
> >
>
> Let the input graph be $G=(V,E)$. The overall training complexity is dominated by the encoder backbone, both GINE and GraphGPS scaling linearly as $O(|V| + |E|)$. Below we summarize the additional asymptotic cost introduced by different masking strategies.
>
> ### Masking distribution
>
> - GraphMAE. Random indices are drawn uniformly and masked via vectorized assignment. The overhead is $O(|V|)$. keeping the total complexity $O(|V| + |E|)$ .
> - StructMAE-P. PageRank computation adds $O(T|E|)$ (with T iterations), followed by two per-graph top-k selections $O(|V|\log|V|)$. The per-batch complexity is $O(|V|\log|V| + |E|)$.
> - StructMAE-L. Replaces PageRank with an MLP+GNN scorer $O(|V|+|E|)$ and the same two top-k operations, yielding the same asymptotic cost $O(|V|\log|V| + |E|)$.
>
> All methods perform masking online without pre-computation, so space complexity remains $O(|V|+|E|)$.
>
> **Summary:** Non-uniform masking adds $O(|V|\log|V|)$ overhead for node scoring and sorting, explaining the observed 2–3× slowdown relative to uniform masking.
>
> ### Prediction target
>
> - **AttrMask.** Predicts atom element types with a simple MLP decoder. Time and space complexity remain $O(|V| + |E|)$.
> - **MotifPred.** Predicts motif (fragment) classes. Each graph is pre-fragmented into motifs $M$, and a padded motif–atom index table of shape $|M|\times L$ ($L=$ max number of atoms in a motif) enables direct indexing for masking. Mask sampling adds only $O(|M|\cdot L)=O(|V|)$, so overall time complexity stays $O(|V| + |E|)$. The extra space $O(|M|\cdot L)$ is minor since $|M|, L \ll |V|$.
>
> ## 2. Interpretation of the non-uniform masking assumption
>
> > The hypothesis that non-uniform masking distributions would outperform uniform sampling wasn't supported by the results. A deeper analysis of why these strategies fail to deliver benefits could be insightful.
> >
>
> We would like to clarify that we **did not hypothesize ourselves** that non-uniform masking should outperform uniform sampling. Rather, our point is that prior works proposing non-uniform distributions implicitly rely on a **necessary assumption**: if a non-uniform masking distribution truly yields a stronger pre-training signal, then this should be reflected in **a stronger dependence of pretrain and downstream labels**.
>
> We note that the **MI analysis in Sec. 5.1** was designed precisely to analyze why non-uniform masking  fails: by isolating $P_\mathcal{M}(X)$ and measuring $I(X;Y)$ model-agnostically, we show the necessary condition is not met in practice.
>
> The results show that this necessary condition is not met: MI values across distributions are nearly identical, which helps explain why their downstream performance also converges.
>
> We acknowledge the wording can be improved and have re-organized relevant sentences to Sec. 3.2.
>
> ## 3. Importance of edge type
>
> > ...Under what conditions or types of tasks might edge masking be more beneficial?
> >
>
> We confirm that edge masking brought no clear benefit in our 2D molecular setup, where only four bond types are predicted from atom embeddings. As atom types and connectivity already determine bonds almost deterministically, masking edges adds little information. However, it may be more effective in settings with richer edge features (e.g., stereochemistry, 3D geometry) or when explicitly modeling edges as first-class tokens.
>
> ### 4. Additional reference
> > The article mainly discusses masking strategy. It is recommended to introduce...
> >
> We appreciate the reviewer’s suggestion. These works explore complementary directions. We have added them in Sec. 2 to acknowledge these developments.
>
> ### 5. Decoder
> > The impact of using a more expressive GNN-based decoder versus a simple MLP decoder was secondary to hyperparameter tuning. Is a more detailed comparison of different decoder architectures needed to clarify their respective roles?
> >
> The decoder choice had limited impact in our setting, likely because the prediction targets are low-dimensional and do not require expressive decoders. More complex reconstruction tasks might benefit from stronger decoders, but this has not been explicitly verified. We therefore consider a detailed comparison of decoder architectures an interesting direction for future investigation.
>
> | Section | Modification | Purpose |
> | --- | --- | --- |
> | App. A.3.1 | Added analysis of time complexity | Explain the slow down of changing masking distribution |
> | Sec. 2 | Included additional recent works | Acknowledge recent advancement utilizing LLMs |
> | Sec. 3.2 | Clarified necessary motivation | Avoid misinterpreting claims |

---

### Review · Reviewer_gnVz · 2025-09-22

**Summary Of Contributions:**

This paper introduces a unified framework for masked pre-training of self-supervised molecular embedding models in the context of molecular property prediction. The frameworks allows flexibility on a few key dimensions relevant to masked pre-training, and the authors analyse the impact of various instantiations of their framework on a number of downstream molecular property prediction tasks by fine-tuning the pre-trained models. The authors also introduce provide some mathematical analysis on the importance of some of these design decisions, such as methods for selecting how the masking is performed and for selecting what to use as a pre-training prediction target.

**Additional Comments:**

## Questions

- In figure 5 the mutual information for the Motif task is significantly larger than other tasks. Couldn't this just be due to the larger size of these fragments? I assume the MI could also be influenced by the masking rate which, as mentioned above, is not controlled for in different setups.

**Audience:**

Yes

**Audience Explanation:**

Pre-trained molecular embedding models are a very active area of research and would be of significant interest for various applications beyond molecular property prediction. This paper attempts to create a unified framework for studying how various design decisions within the pre-training setup can influence the downstream performance of pre-trained models for molecules.

**Claims And Evidence:**

No

**Claims Explanation:**

- The setup of the pre-training framework seems mostly sound. They are a number of caveats, however. Firstly, the masking rates of various models used is not the same, and no attempt is made to investigate the importance of the amount of masking on the downstream performance. Without making an attempt to deconvolve the impact of this hyperparameter on the others, it's difficult to draw strong conclusions about the importance of the hyperparameters which are under investigation. Secondly, some modifications are made to the original models to make them fit into the framework, but it's difficult to see how much of an impact these changes make to the performance. For example, for the MotifPred-L (T) model, this involves using Gumbel-softmax to ensure differentiability, but this is known to have the potential of causing stability issues during training and may be detrimental to performance. It's therefore difficult to see whether the conclusions in the paper would also hold for the original models.

- The pre-training dataset used is much smaller than many datasets used previously for pre-training molecular models. This doesn't mean the paper is wrong necessarily, but makes it difficult to see how the claims of the paper would transfer to more commonly used settings. The small model sizes used in this study are a related concern.

- One of the core contributions of the paper is a mathematical analysis of the mutual information between the pre-training target, X, and the fine-tuning targets, Y (molecular properties). The basis of their analysis seems puzzling to me, however. They claim  (second last paragraph of page 5) "This hypothesis implies that: if a non uniform strategy $\mathcal{M}$ is indeed more effective, the information provided to $Y$ given $X \sim P_{\mathcal{M}} (. | G)$ should be higher than that given $X \sim P_{\text{uniform}} (. | G)$". I don't think this holds however (and since it forms the core of your argument for conducting this analysis it would be very helpful to write this more formally). One can maximise the information in X by masking everything but this is not necessarily a useful pre-training target. Since pre-training and fine-tuning are done in separate phases a better question to ask is how useful are the weights of the model for fine-tuning immediately after pre-training. At the moment I am not convinced the mutual information results presented here are particularly meaningful.

**Requested Changes:**

- As mentioned above, a more thorough and formal explanation of why you have chosen to analyse the mutual information between X and Y.

- The paper also currently does not include any discussion or analysis of models which encode edge information and apply masking to edges (such as [1] and others), which are state of the art for many of these tasks. It's not necessary to include these models within the framework but the paper should at a minimum make clear that techniques such as edge attention or masking is an important part of the discussion that is missing from the current framework.

- Figure 3 should probably include error bars

- In the section comparing GIN with GraphGPS, it would be useful to discuss whether there is a difference in sampling time or compute resource between these architectures. Could the benefit of GraphGPS simply be due to more compute?

- Overall, I also think the introduction, background work, and framework sections could be significantly reduced in size. The first approx. 11 pages of the paper are devoted to outlining the details of the framework that is then used to test hyper parameters of existing model setups. A lot of this is very verbose and doesn't add much to the paper.  I think a lot of this could be condensed and more in depth details moved to the appendix. But I ultimately leave it up to the AE to decide whether this should be an extended length paper or not.

[1] "An end-to-end attention-based approach for learning on graphs" https://arxiv.org/abs/2402.10793

---

> ### Author Response · Authors · 2025-10-04
> **Author Response to Reviewer gnVz (Part I)**
>
> We thank the reviewer for their constructive comments. Below we summarize our responses in order of importance. All additional experiments were run with 5 random seeds, and a summary table of manuscript revisions is provided at the end.
>
> ## 1. Mutual Information analysis
>
> ### Reviewer comment (summary):
> > Questioned the validity of our MI formulation and stated one can maximize information of X by masking everything.
> >
>
> We first clarify that in our framework, $X$ and $Y$ are formulated as random variables, defined on the disjoint union of $\mathcal{S}(G)$. MI depends solely on the **distribution** of $X,Y$ and is independent of the mask rate. Second, the “mask everything” case does not maximize MI. This will be undesirable because input graphs becomes uninformative when fully masked, not our definition of $X$ ceases to be well-defined.
>
> Importantly, we do not claim higher MI is sufficient for better downstream performance. Rather, we test the *necessary* assumption underlying prior non-uniform masking works: if a non-uniform distribution truly provides a stronger pretraining signal, $I(X;Y)$ should increase. Our results show this condition is not met, explaining why non-uniform masking yields no benefit. Wording in Sec. 3.1–3.2 has been revised for clarity.
>
> > how useful are the weights ... after pre-training
> >
> We further evaluated the **frozen-backbone** setting (encoder fixed, only a linear classifier trained):
> | Method | GraphMAE | StructMAE-P | StructMAE-L |
> | --- | --- | --- | --- |
> | Avg. ROC-AUC | 65.9 (0.7) | 64.8 (0.2) | 63.8 (0.4) |
>
> | Method | AttrMask | MotifPred | AttrMask (T) | MotifPred (T) |
> | --- | --- | --- | --- | --- |
> | Avg. ROC-AUC | 62.6 (0.6) | 63.4 (0.7) | 63.3 (0.1) | **69.0 (0.6)** |
>
> The results confirm our main conclusion. Per-dataset results are given in App. A.1.2.
>
> ## 2. Masking rate alignment
> > “The masking rates of various models used is not the same”
> >
>
> We did unify the mask rates within each group of comparison, based on the mask ratio used in the original framework (i.e. 0.15 for AttrMask and 0.25 for GraphMAE). We have included a table explicitly listing that in Sec. 4 for clarity.
>
> > “and no attempt is made to investigate the importance of the amount of masking…”
> >
> We added following results on average classification performance under additional mask ratios (0.10, 0.15, 0.35). It again shows that changing the masking distribution does not yield noticeable differences, supporting our first conclusion.
>
> | method \ mask_rate | 0.1 | 0.15 | 0.25 | 0.35 |
> | --- | --- | --- | --- | --- |
> | GraphMAE | 71.5 (0.2) | 71.8 (0.4) | 71 (0.4) | 72.2 (0.6) |
> | StructMAE-P | 71.4 (0.4) | 72.0 (0.4)  | 71.1 (0.5) | 71.3 (0.3)  |
> | StructMAE-L | 72.2 (0.4) | 71.6 (0.5) | 70.6 (0.4) | 72.1 (1.6) |
>
> We note that **Appendix A.2 (Fig. 8)** already reports a mask-ratio sensitivity sweep for AttrMask and MotifPred on both encoders over {0.10,0.15,0.25,0.40,0.50}. We acknowledge that the description in A.2 can be clearer and have revised the text.
>
> ## 3. Motif MI is not explained by vocabulary size or mask rate
>
> > *“… motif MI is significantly larger—could this just be due to larger fragment size or masking rate?”*
> >
>
> As clarified in response 1, MI does not depend on the mask rate.
>
> Regarding vocabulary size: although motif labels can involve more categories, this factor alone does not account for the much higher MI observed. For example, in the Bace dataset, the motif label set has size 407, which is smaller than the 512 codebook labels used in MoleBERT (MAM-VQ/A), yet the motif MI is still significantly larger.
>
> To further test this, we conducted a shuffle control: motif labels X were randomly reassigned across positive and negative molecules ($Y=1,0$), preserving the vocabulary size but destroying semantic alignment (see A.6.2 in the revised manuscript). The shuffled MI collapsed to the level of atom-label baselines, showing that a larger label range is not the mere reason of higher MI. This confirms that the observed advantage of motif labels reflects genuine statistical dependence with $Y$, rather than an artifact of the number of classes.
>
> ## 4. Modifications made to the original models
>
> > *“Secondly, ... difficult to see whether the conclusions in the paper would also hold for the original models.”*
> >
>
> Our goal is not to reproduce every original implementation, but to isolate the effects of masking distribution and prediction target under a unified framework.
>
> Example: MoleBERT (MAM-A/VQ) originally combines masked prediction with an extra contrastive loss. We retain only its Argmax/VQ tokenization while aligning the rest with the AttrMask setup. Similarly, MotifPred-L (T) adopts Gumbel-softmax for differentiable masking. While we did not observe instability, our conclusions are unaffected by the specific implementation.
>
> We have added a summary table listing key adaptations.
>
> Due to length constraints, we provide additional clarifications in a separate comment below.

---

> ### Author Response · Authors · 2025-10-04
> **Author Response to Reviewer gnVz (Part II)**
>
> ## 5. Dataset and model scale
>
> **Reviewer comment:**
>
> > *“The pre-training dataset used is much smaller… makes it difficult to see how the claims transfer to larger-scale settings. The small model sizes are a related concern.”*
> >
>
> **Our response:**
>
> We acknowledge this limitation. Our study was designed as a controlled investigation to isolate the effects of the three design dimensions, rather than to achieve state-of-the-art performance. Larger-scale pretraining and models are a natural next step, and we view our work as providing a principled foundation and methodological clarity for such future studies.
>
> ---
>
> ## 6. GraphGPS vs. GIN compute fairness
>
> **Reviewer comment:**
>
> > *“... it would be useful to discuss whether there is a difference in sampling time ... Could the benefit of GraphGPS simply be due to more compute?”*
> >
>
> **Our response:**
>
> We already report time comparisons in A.3. As shown there, wall-clock training times of two encoders are comparable under matched configurations. Frozen-backbone results in Response 1 provide direct evidence that the gains are not due to compute alone:
>
> - AttrMask remains weak on GPS, despite using the same encoder resources.
> - MotifPred on GIN also performs poorly.
> - Significant improvements only emerge for the (MotifPred + GPS) combination.
>
> If gains were purely from compute effect, we would expect large gaps for node-level masking as well, which we did not observe in general. Also see A.1.1 Table 9.
>
> ---
>
> ## 7. Edge masking not discussed
>
> **Reviewer comment:**
>
> > *“The paper does not include models which encode edge information… should at a minimum acknowledge.”*
> >
>
> **Our response:**
>
> We agree edge-aware methods are relevant. We did include edge-masking baselines (e.g., AttrMask-B, MAM-A-B; Table 8), but they showed no gains in general. Since the bond attribute is a single-valued label with only 4 classes, this is expected. We thank the reviewer for pointing out the additional reference, and have incorporated a discussion of it in the revision.
>
> ---
>
> ## 8. Figure 3 error bars
>
> **Reviewer comment:**
>
> > *“Figure 3 should probably include error bars.”*
> >
>
> **Our response:**
>
> We agree and have added error bars across multiple seeds to the revised figure.
>
> ---
>
> ## 9. Verbosity of introduction/framework
>
> **Reviewer comment:**
>
> > *“… the first ~11 pages are very verbose… should be condensed.”*
> >
>
> **Our response:**
>
> Our intention was to keep the Section 3 (Methodology) self-contained, since many readers may not be familiar with the specific implementations of prior methods, and we wanted to avoid forcing them to constantly switch to the appendix or original references. That said, we have streamlined some passages to reduce redundancy while retaining the essential details in the main text for clarity and accessibility. As a result, the main text of the revised version has been condensed from 20 to 18 pages.
>
> ## Summary of revision
>
> | Section | Modification | Purpose |
> | --- | --- | --- |
> | Sec. 3.1–3.2 | Clarified necessary (not sufficient) motivation | Avoid overstated claims |
> | App. A.1.2 | Added frozen-backbone results | Address overstated gains |
> | App. A.2 | Added another mask-ratio sensitivity study | Reinforce conclusion on mask distribution |
> | App. A.6.2 | Shuffled-label sanity check | Show motif MI advantage is semantic |
> | App. A.10, A.13 | Moved description of addtional variants from Sec. 5.1; added a table of key adaptations made from previous works | Improve clarity |
> | Sec. 2.1 | Added discussion on edge-centric method | Complementary to related works |
> | Fig. 3 | Added error bars, rescaled y-axis | Improve visualization |
> | Sec. 3.3, 4 & App A.12 | Move full algorithms of StructMAE to A.12; streamlined Sec.4; added a categorization table with key elements in examined methods | For clarity and conciseness |

---

> > ### Comment · Reviewer_gnVz · 2025-10-24
> >
> > Thank you for your extensive rebuttal and additional experiments! You have certainly addressed many of my concerns and I am very happy with the updated paper.
> >
> > One final point - I believe some of my questions were indeed answered in the Appendix of the original paper, but I missed the pointers to these in the main text. I think the updates to the paper have improved this, but it would be worth checking to ensure that these important results in the Appendix are not missed by including references in the appropriate parts of the main text.

---

> > > ### Author Response · Authors · 2025-10-25
> > >
> > > Thank you for the suggestion! We are currently checking the main text and will incorporate the references together with any remaining feedback from other reviewers before uploading the final revised version.

---

### Decision · Action_Editor_yvxG · 2025-11-02

**Recommendation:** Accept as is

**Additional Comments:**

N.A.

**Audience:**

Yes

**Audience Explanation:**

This work systematically investigates core design choices in pre-trained molecular models, which is an active area of research. The comprehensive analysis provided will be of clear interest to TMLR readers working in GNN research and its applications such as computational chemistry.

**Claims And Evidence:**

Yes

**Claims Explanation:**

This paper proposes a unified framework to systematically investigate the impact of three key components in masking-based pre-training for molecular graphs: masking distributions, prediction targets, and encoder architectures.

The paper’s main claims are as follows:
1. The influence of the masking distribution is less significant than often assumed.
2. The choice of the prediction target, particularly a motif-based target, has a substantial impact on the model's performance on downstream tasks.
3. The importance of the encoder architecture is conditional: a more expressive architecture becomes effective when the prediction target itself has rich semantics.

To support these claims, the authors also introduce a novel evaluation methodology based on Mutual Information and Jensen-Shannon Divergence.

Three expert reviewers reviewed this manuscript. The initial reviews raised valid questions. Some reviewers noted that the original claims felt overstated and that addressing this would improve the paper's quality. The authors provided an exemplary rebuttal and a thoroughly revised manuscript that directly addresses these concerns.

Based on the authors' revisions and responses, all reviewers now concur that the paper's claims are well-supported by the evidence. While the initial concerns about overstatement were valid, the authors’ revisions and the resulting consensus among the reviewers confirm that these issues do not constitute grounds for rejection.

Therefore, I assess that the paper's claims are now adequately supported by the evidence.